# BRIDGEV2W: BRIDGING VIDEO GENERATION MODELS TO EMBODIED WORLD MODELS VIA EMBODIMENT MASKS

## ABSTRACT

Embodied world models have emerged as a promising paradigm in robotics, most of which leverage large-scale Internet videos or pretrained video generation models to enrich visual and motion priors. However, they still face key challenges: a misalignment between coordinate-space actions and pixel-space videos, sensitivity to camera viewpoint, and non-unified architectures across embodiments. To this end, we present **BridgeV2W**, which converts coordinate-space actions into pixel-aligned embodiment masks rendered from the URDF and camera parameters. These masks are then injected into a pretrained video generation model via a ControlNet-style pathway, which aligns the action control signals with predicted videos, adds view-specific conditioning to accommodate camera viewpoints, and yields a unified world model architecture across embodiments. To mitigate overfitting to static backgrounds, BridgeV2W further introduces a flow-based motion loss that focuses on learning dynamic and task-relevant regions. Experiments on single-arm (DROID) and dual-arm (AgiBot-G1) datasets, covering diverse and challenging conditions with unseen viewpoints and scenes, show that BridgeV2W improves video generation quality compared to prior state-of-the-art methods. We further demonstrate the potential of BridgeV2W on downstream real-world tasks, including policy evaluation and goal-conditioned planning.

## 1 INTRODUCTION

Embodied world models have gained increasing attention in the robotics community for their ability to model the physical dynamics of the environment. They support policy evaluation by simulating counterfactual action outcomes (Jiang et al., 2025; Li et al., 2025d; 1X World Model Team, 2025) and goal-conditioned planning by forecasting future states toward specified goal images (Assran et al., 2025; Bar et al., 2025; Baldassarre et al., 2025; Bai et al., 2025). Despite this potential, current embodied world models remain constrained by limited task-relevant data and mismatched design (i.e., coordinate-space actions misaligned with the pixel-space videos).

Current research on action-conditioning embodied world models generally follows two main paradigms. The first trains models from scratch using only domain-specific robot data. While effective within the same domain, such models often fail to generalize well to unseen scenarios. The second paradigm leverages large-scale Internet videos to incorporate rich visual and motion priors, either by directly pretraining a model or by fine-tuning a pretrained video generation model with action conditioning. Although notable progress has been made, these approaches still suffer from several key limitations, as outlined in Figure 1:

(1) **Action-Video Gap**: Most action-conditioning methods represent the end-effector poses as coordinate-space actions that lie in a low-dimensional geometric space, whereas pretrained video generation models operate in a high-dimensional pixel space. This representation space mismatch weakens conditioning and limits the reuse of pretrained visual and motion priors. (2) **Viewpoint Sensitivity**: Coordinate-space actions are highly sensitive to camera viewpoint changes. Even for the same action, existing methods still struggle to generate reasonable future states when the camera viewpoint varies, limiting their applicability in unseen camera viewpoint settings. (3) **Non-Unified Architecture Across Embodiments**: These methods lack a unified architecture across robot em-

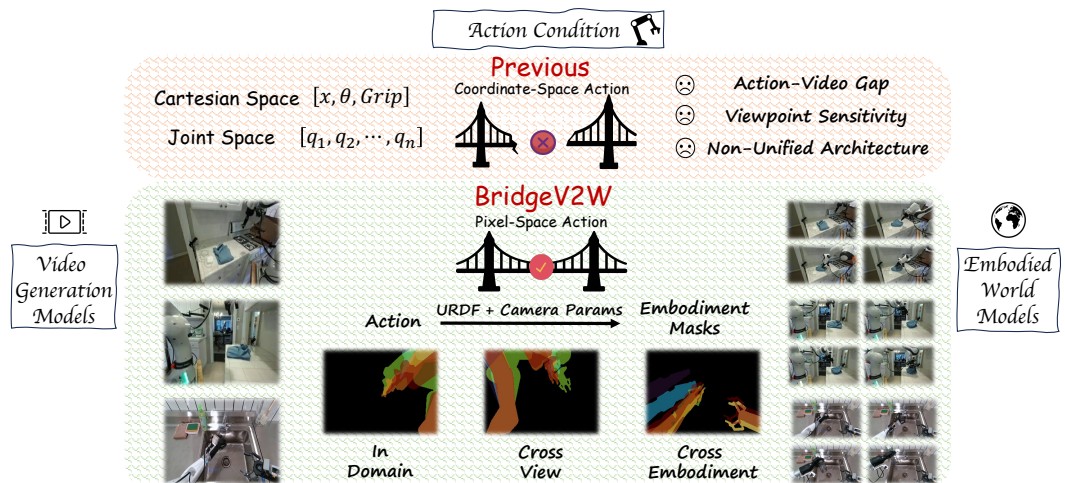

Figure 1: BridgeV2W vs. previous methods. Pixel-aligned embodiment masks bridge video generation models to embodied world models, addressing the action–video gap, improving viewpoint robustness, and yielding a unified architecture across embodiments.

bodiments. For instance, single-arm and dual-arm systems have different degrees of freedom and typically require separate action encoders, which limits knowledge transfer across embodiments and hinders the scalability required for building truly general-purpose embodied world models.

We start from a simple observation: **if we transform the action representation into a pixel-aligned mask that reflects the embodiment's actual motion, the issues mentioned above can be substantially mitigated.** In particular, projecting actions into pixel-aligned masks (i) closes the gap to pixel-space of pretrained video generation models because *the conditioning also lies in pixel space*, (ii) facilitates view-conditioned robustness across diverse camera viewpoints because *supervision is anchored in the image plane*, and (iii) unifies action conditioning across embodiments because *the mask is agnostic to robot-specific action space*.

To this end, we propose **BridgeV2W**, a framework that bridges pretrained video generation models to embodied world models via embodiment masks. We leverage the robot's Unified Robot Description Format (URDF), which is easy to obtain and widely available, together with the camera's intrinsic and extrinsic parameters to render actions into view-specific embodiment masks. These masks encode the embodiment's structure in pixel space, providing a spatial conditioning signal to pretrained video generation models. Inspired by ControlNet-style conditioning (Zhang et al., 2023), we inject these masks as spatial conditions into the pretrained video generation models, enabling action-conditioned generation while preserving their pretrained visual and motion priors. These pixel-aligned mask actions match the pixel space of the pretrained video generation model, preserve viewpoint-specific variation, and enable a unified world model architecture across embodiments.

While the above design already enables BridgeV2W to provide effective action conditioning, conventional frame-level reconstruction losses reconstruct all regions without distinction, including most task-irrelevant ones. For robotic manipulation, it is more effective to focus on motion regions, including the embodiment and manipulated objects. Thus, we further introduce a flow-based motion loss that computes optical flow between predicted and ground-truth frames. By penalizing discrepancies only in motion regions, BridgeV2W focuses more on task-relevant motion patterns rather than static background details.

Our main contributions can be summarized as follows:

- We present BridgeV2W, a framework that bridges pretrained video generation models to embodied world models via embodiment masks. Using a ControlNet-style conditioning mechanism, BridgeV2W preserves pretrained visual and motion priors while enabling viewpoint robustness and a unified architecture across embodiments.

- We introduce a flow-based motion loss that encourages the model learning toward task-relevant areas of the embodiment and manipulated objects, reducing emphasis on static background reconstruction.

- We evaluate BridgeV2W on single-arm (DROID (Khazatsky et al., 2024)) and dual-arm (AgiBot-G1 (Bu et al., 2025)) datasets, covering unseen viewpoints and scenes, and demonstrate enhanced generation quality compared to previous state-of-the-art methods. We further validate BridgeV2W on downstream robotic tasks: as a proxy for policy evaluation, it shows strong correlation with real-world success, and as a goal-image–conditioned planner, it achieves promising manipulation performance.

## 2 RELATED WORKS

### 2.1 LEARNING EMBODIED WORLD MODELS VIA VIDEO GENERATION

Existing approaches that adopt video generation models as embodied world models can be broadly divided into two categories. The first directly trains a video generation model from scratch using domain-specific robot data (Jiang et al., 2025; Liao et al., 2025; Bai et al., 2025; Wang et al., 2025b; Zhu et al., 2024; Wu et al., 2024), learning environment dynamics purely from task demonstrations. While such models can accurately capture the physics and visual patterns of the training domain, their reliance on limited and task-specific data often leads to overfitting, resulting in poor generalization to unseen scenarios.

The second category leverages large-scale Internet videos or pretrained video generation models to incorporate rich visual and motion priors (He et al., 2025; Agarwal et al., 2025; Assran et al., 2025; 1X World Model Team, 2025; Huang et al., 2025). A common strategy is to fine-tune these models by introducing action conditioning. This paradigm benefits from the diversity and scale of pretraining data, making it more adaptable to varied visual scenes. However, existing methods still suffer from several limitations, including the mismatch between coordinate-space actions and pixel-space videos, sensitivity to camera viewpoint changes, and the lack of a unified architecture for different robot embodiments. To address these challenges, we propose BridgeV2W, which integrates embodiment masks into a pretrained video generation model via a ControlNet-style conditioning mechanism, enabling robust multi-view generation, architecture sharing across embodiments, and better alignment between action inputs and the model's visual priors.

### 2.2 INJECTING CONDITIONS INTO PRETRAINED GENERATION MODELS

ControlNet (Zhang et al., 2023) extends diffusion-based generative models (Rombach et al., 2022) by introducing additional condition-encoding branches that start with zero-initialized parameters. This design ensures that the pretrained network's behavior remains unchanged at the beginning of fine-tuning, while gradually learning to incorporate new guidance signals such as sketches, depth maps, or human poses. In both image (Li et al., 2024) and video generation (Guo et al., 2024; Wang et al., 2024b; Bar-Tal et al., 2024), prior works employ ControlNet-style conditioning to guide generation with diverse control signals, including optical flow (Li et al., 2025c; Shi et al., 2024; Zhang et al., 2025; Yin et al., 2023), bounding boxes (Li et al., 2025a; Namekata et al., 2025; Qiu et al., 2024; Wang et al., 2024a), and point trajectories (Fu et al., 2024; Gu et al., 2025; Wang et al., 2025a;c). In BridgeV2W, we adapt this conditioning strategy to inject embodiment masks into a pretrained video generation model, and we further introduce flow-based loss to focus more on task-relevant areas, including embodiments and manipulated objects.

## 3 BRIDGEV2W FRAMEWORK

### 3.1 DEFINITION OF EMBODIED WORLD MODELS

The term *world model* has been broadly discussed in previous works (e.g., latent dynamic predictors or pure video generators without actions). In this paper, we adopt an operational, action-conditioned definition of embodied world models that requires *an initial frame and an action sequence as inputs* and *an RGB video as output*. Given an initial RGB frame $I_0 \in \mathbb{R}^{H \times W \times 3}$ and an action sequence

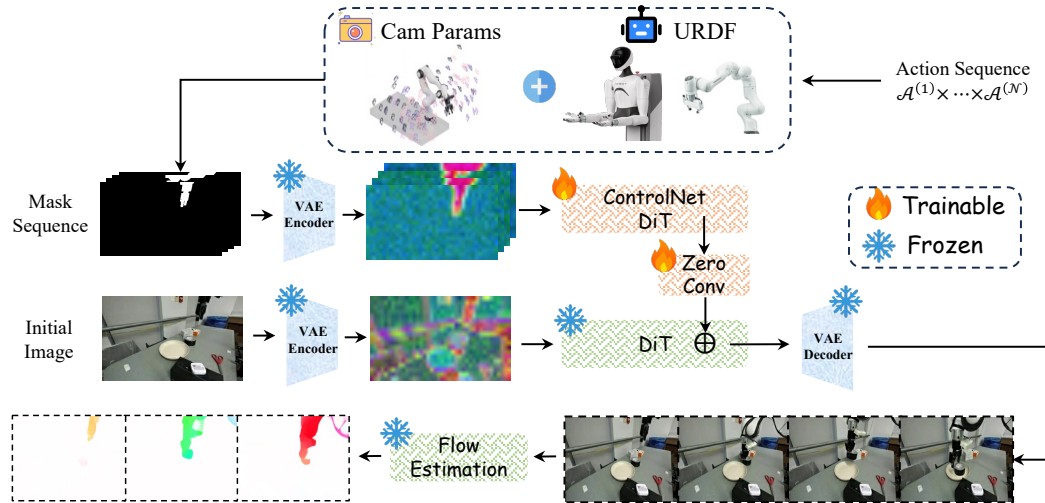

Figure 2: Overview of the BridgeV2W pipeline. Actions are projected into pixel-space masks using URDF and camera parameters. The initial image and mask sequence are encoded by VAE, with mask features injected via a ControlNet branch into the DiT backbone. The model generates action-consistent videos, trained with diffusion, dynamics-consistency, and flow-based objectives.

$\mathbf{a}_{0:T-1}$, the model $f_\theta$ predicts the future frames

$$\hat{V}_{1:T} \;=\; f_\theta(I_0, \mathbf{a}_{0:T-1}) \tag{1}$$

where $T$ is the video length and $V \triangleq I_{0:T}$. The action space $\mathcal{A}$ is embodiment- and parameterization-agnostic: we allow $N$ manipulators with $\mathcal{A} \triangleq \mathcal{A}^{(1)} \times \cdots \times \mathcal{A}^{(N)}$ and $a_t \triangleq [a_t^{(1)}; \ldots; a_t^{(N)}]$, where each manipulator's action $a_t^{(n)}$ may be specified either in Cartesian space, $a_t^{(n)} = (p_t^{(n)}, R_t^{(n)}, g_t^{(n)})$ with $p_t^{(n)} \in \mathbb{R}^3$, $R_t^{(n)} \in \mathrm{SO}(3)$, and gripper control $g_t^{(n)}$; or in joint space, $a_t^{(n)} = q_t^{(n)} \in \mathbb{R}^{d_n}$, with $d_n$ the number of actuated joints. Our definition does not prescribe a particular architecture or training objective; thus, any model that maps an initial frame and an action sequence to the corresponding RGB video qualifies as an embodied world model under this definition.

## 3.2 EMBODIMENT MASK EXTRACTION FROM ACTION SEQUENCES

To extract embodiment masks from action sequences, we use a URDF description of the embodiment and simulate its motion via forward dynamics to recover its 3D structure. With known camera intrinsics and extrinsics, the embodiment is projected onto the image plane, yielding a per-frame mask $m_t$ from the camera's viewpoint. We align masks to the predicted frames and denote the mask sequence as $M \triangleq \{m_t\}_{t=1}^T$. This ensures that each action $a_t$ is consistently paired with its associated mask $m_t$, producing tuples $(I_0, M, V_{1:T})$.

The pipeline is extensible to settings without explicit action annotations or camera calibration, e.g., human–hand interaction videos (Goyal et al., 2017; Grauman et al., 2022). In such cases, segmentation tools like GroundedSAM (Ren et al., 2024) can be used to directly extract $M$ from raw video, also enabling datasets of the form $(I_0, M, V_{1:T})$. This highlights the flexibility of our framework and its ability to integrate embodiment signals from diverse sources. Additional details are provided in Appendix B.5.

## 3.3 CONTROLNET-BASED VIDEO GENERATION

The overall pipeline of BridgeV2W is illustrated in Figure 2. We adopt CogVideoX-5B-I2V (Yang et al., 2025c) as the image-to-video backbone, which utilizes 3D full attention to capture high-quality temporal dynamics. Given an input image $I_0 \in \mathbb{R}^{H \times W \times 3}$ and a target video $V \in \mathbb{R}^{T \times H \times W \times 3}$, a

pretrained 3D VAE (Kingma & Welling, 2013) encodes them into latent tensors

$$z^{img}, z^{vid} \in \mathbb{R}^{T_\ell \times H_\ell \times W_\ell \times C}, \quad T_\ell = \frac{T}{4}, \; H_\ell = \frac{H}{8}, \; W_\ell = \frac{W}{8}, \; C = 16$$

The image latent $z^{img}$ is zero-padded along the temporal axis to length $T_\ell$ and concatenated with a noised version of the video latent tensors (see diffusion below). A Diffusion Transformer (DiT) (Peebles & Xie, 2023) progressively denoises the latent tensors and yields the output $\hat{V}_{1:T} \in \mathbb{R}^{T \times H \times W \times 3}$ after the VAE decoding process.

We introduce a mask-sequence ControlNet conditioned on pixel-aligned masks $M \in \mathbb{R}^{T \times H \times W \times 1}$. Following ControlNet (Zhang et al., 2023), the mask sequence is encoded by the same 3D VAE into

$$z^{mask} \in \mathbb{R}^{T_\ell \times H_\ell \times W_\ell \times C}$$

and injected via a set of trainable DiT blocks. Each ControlNet block produces features that pass through zero-initialized convolutional layers and are additively fused into the corresponding backbone DiT block, guiding generation to align with the action-to-mask correspondence.

Following (Yang et al., 2025c), we adopt velocity prediction in latent space. Let $z_0$ denote the clean video latents and $\epsilon \sim \mathcal{N}(0, I)$ be Gaussian noise. At diffusion step $\tau$,

$$\tilde{z}_\tau \;=\; \sqrt{\alpha_\tau}\, z_0 \;+\; \sqrt{1 - \alpha_\tau}\, \epsilon \tag{2}$$

where $\alpha_\tau$ is the noise-schedule coefficient. The model predicts the velocity $v_\theta$, and the training objective is:

$$\mathcal{L}_{diff} = \mathbb{E}_{\tau,\, \epsilon \sim \mathcal{N}(0,I),\, z_0} \Big[ \| z_0 - (\sqrt{\alpha_\tau}\, \tilde{z}_\tau - \sqrt{1 - \alpha_\tau}\, v_\theta) \|_2^2 \Big] \tag{3}$$

While the standard video diffusion loss (Eq. 3) supervises frames independently, it under-exploits temporal correlations and can degrade spatiotemporal coherence. To mitigate this, we adopt a dynamics-consistency objective from Yang et al. (2025b) that explicitly supervises latent motion, i.e., differences between video latents across temporal offsets. Let $\{z_t\}_{t=0}^{T_\ell}$ denote the ground-truth video latents (from the VAE) and $\{\hat{z}_t\}_{t=0}^{T_\ell}$ the model-predicted latents in the same latent grid. The dynamics-consistency loss is:

$$\mathcal{L}_{dyn} \;=\; \sum_{j=1}^{K} \frac{1}{T_\ell - j} \sum_{t=0}^{T_\ell - 1 - j} \| (\hat{z}_{t+j} - \hat{z}_t) - (z_{t+j} - z_t) \|_2^2 \tag{4}$$

where $K$ is the maximum temporal offset (set to $4$ in our experiments) and $T_\ell$ is the number of latent frames. By computing this objective over multiple offsets $j = 1, \ldots, K$, it captures both short and long horizon dynamics. The total training loss combines the frame-wise diffusion objective and the dynamics-consistency term:

$$\mathcal{L} \;=\; \mathcal{L}_{diff} \;+\; \lambda_{dyn}\, \mathcal{L}_{dyn}$$

### 3.4 FLOW-LOSS ENHANCED MODULE

While embodiment-aware mask action conditioning helps, conventional frame-level reconstruction losses still encourage uniform fidelity over entire frames, including task-irrelevant backgrounds. For robotic manipulation, supervision should prioritize the motion of the embodiment and manipulated objects. We therefore adopt an optical-flow-based objective that compares the motion fields of the predicted and ground-truth videos, emphasizing dynamic, task-relevant regions and further improving spatiotemporal coherence.

Let $\hat{V}_{1:T}$ denote the predicted video (decoded from the VAE latents) and $V_{1:T}$ the ground-truth video. We use a pretrained, frozen RAFT flow estimator (Teed & Deng, 2020) to compute optical flows for both sequences. The flow discrepancy aggregates a direction term (cosine-based) and a magnitude term (Huber-based (Huber, 1992)) to focus on learning motion regions. In compact form,

$$\mathcal{L}_{flow} \;=\; Loss\big(F_\phi(\hat{V}_{1:T}), F_\phi(V_{1:T})\big) \tag{5}$$

where $F_\phi(\cdot)$ is the frozen flow operator and $Loss\,(\cdot, \cdot)$ denotes the composite direction–magnitude discrepancy.

The overall training objective augments the diffusion loss and the latent dynamics-consistency term with flow supervision:

$$\mathcal{L}_{total} = \mathcal{L}_{diff} + \lambda_{dyn}\,\mathcal{L}_{dyn} + \lambda_{flow}\,\mathcal{L}_{flow} \tag{6}$$

To ensure stability when video quality is poor at the beginning of training, the flow loss is disabled during an initial warm-up of $e_{switch}$ epochs and then activated at a fixed weight:

$$\lambda_{flow} = \begin{cases} 0, & e < e_{switch} \\ \lambda_{flow}^{\star}, & e \geq e_{switch} \end{cases} \tag{7}$$

All flow computations are performed in the pixel domain on VAE-decoded frames, ensuring the penalty directly reflects the perceptual motion of the embodiment and manipulated objects.

## 4 EXPERIMENTS

In this section, we perform experiments on two robotic datasets as well as in the real world to evaluate the proposed BridgeV2W. Through the experiments, we aim to answer five research questions:

**RQ1:** Is representing actions as pixel-aligned embodiment masks more compatible with pretrained video generation models than coordinate-space action representations?

**RQ2:** Can BridgeV2W inherit the visual and motion priors of a pre-trained video generation model and demonstrate generalization in novel scenarios?

**RQ3:** Does embodiment-mask supervision enable robustness to unseen camera viewpoints?

**RQ4:** Can BridgeV2W be deployed on multiple embodiments under a unified world model?

**RQ5:** Can BridgeV2W be applied to downstream tasks, serving as a module for real-world policy evaluation and goal-image-based planning?

### 4.1 EXPERIMENTAL SETUP

**Datasets.** We evaluate BridgeV2W on two robotic datasets: DROID (Khazatsky et al., 2024), featuring a single-arm Franka system, and AgiBot-G1 (Bu et al., 2025), featuring a dual-arm system. For training, we use 19k trajectories from DROID, each recorded from two calibrated cameras, and 15k trajectories from AgiBot-G1. The provided camera calibration parameters are used to generate embodiment masks. For evaluation, we reserve 200 trajectories from each dataset as the standard test set. In addition, for DROID, we further hold out 100 trajectories for *unseen-camera-viewpoint* evaluation and another 100 trajectories for *unseen-scene* evaluation.

**Implementation Details.** We adopt CogVideoX-5B-I2V (Yang et al., 2025c) as the pretrained video generation backbone. All video frames are resized to a resolution of $720 \times 480$, and the model predicts a horizon of 25 future frames. To accelerate training, we randomly sample 20 clips from each video. Since our world model does not require language input, we ignore the original language instructions provided in the datasets. Instead, because CogVideoX requires a textual prompt for conditioning, we use a fixed placeholder prompt, *"Follow this action sequence represented in mask"*, for all training and evaluation. More details could be found in Appendix B.2.

**Baselines.** We compare BridgeV2W with three state-of-the-art embodied world model methods:

- IRASim (Zhu et al., 2024): a diffusion transformer–based trajectory-to-video model with frame-level action conditioning to capture robot–object interactions.
- Cosmos (Agarwal et al., 2025): a world foundation model platform that pretrains on large-scale video data and supports building customized world models through post-training.
- EVAC (Jiang et al., 2025): an action-conditioned world model that incorporates multi-level action injection and ray-map encoding for generating controllable multi-view videos.

**Metrics.** Our evaluation uses four standard video quality metrics: PSNR (Hore & Ziou, 2010), SSIM (Wang et al., 2004), LPIPS (Zhang et al., 2018), and FVD (Ranftl et al., 2020), which measure pixel-level fidelity, perceptual similarity, and temporal realism of generated videos, respectively. To further assess the alignment between generated videos and input actions, we compute Mask-IoU between embodiment regions in generated and ground-truth frames, obtained via Grounded SAM (Ren et al., 2024) with *"robotic arm"* as the textual prompt.

Table 1: Video generation results of BridgeV2W across different DROID datasets variants.

| Dataset Variant | Methods | PSNR ↑ | SSIM ↑ | LPIPS ↓ | FVD ↓ | Mask-IoU ↑ |
|---|---|---|---|---|---|---|
| In Domain | IRASim (Zhu et al., 2024) | 22.11 | 0.846 | 0.119 | 175.7 | 58.0 |
| | Cosmos (Agarwal et al., 2025) | 21.13 | 0.826 | 0.122 | 184.3 | 59.2 |
| | EVAC (Jiang et al., 2025) | 21.97 | **0.877** | 0.124 | 219.8 | 57.4 |
| | BridgeV2W (Ours) | **22.89** | 0.874 | **0.111** | **145.2** | **62.2** |
| Unseen Camera-Viewpoint | IRASim | 18.02 | 0.763 | 0.162 | 415.8 | 45.9 |
| | Cosmos | 19.73 | 0.786 | 0.177 | 303.1 | 48.0 |
| | EVAC | 20.15 | 0.830 | 0.148 | 224.7 | 52.6 |
| | BridgeV2W (Ours) | **20.87** | **0.833** | **0.127** | **191.3** | **55.3** |
| Unseen Scene | IRASim | 16.23 | 0.672 | 0.166 | 583.8 | 32.7 |
| | Cosmos | 19.38 | 0.709 | 0.147 | 412.2 | 37.0 |
| | EVAC | 17.78 | 0.693 | 0.159 | 486.5 | 31.4 |
| | BridgeV2W (Ours) | **19.73** | **0.717** | **0.138** | **362.1** | **44.1** |

Table 2: Video generation results of BridgeV2W on AgiBot-G1 dataset.

| Methods | PSNR ↑ | SSIM ↑ | LPIPS ↓ | FVD ↓ | Mask-IoU ↑ |
|---|---|---|---|---|---|
| IRASim (Zhu et al., 2024) | 23.38 | 0.842 | 0.121 | 144.6 | 55.6 |
| Cosmos (Agarwal et al., 2025) | 22.96 | 0.857 | 0.135 | 239.7 | 54.8 |
| EVAC (Jiang et al., 2025) | 23.64 | 0.858 | 0.117 | 169.4 | 57.9 |
| BridgeV2W (Ours) | **24.49** | **0.868** | **0.102** | **129.5** | **58.3** |

## 4.2 VIDEO GENERATION EVALUATION

**Evaluation on DROID.** Video generation results of BridgeV2W across DROID dataset variants are shown in Table 1. On all in-domain, unseen-viewpoint, and unseen-scene settings, BridgeV2W consistently achieves stronger temporal realism and perceptual quality (lower FVD/LPIPS) and tighter action–video alignment (higher Mask-IoU), while keeping PSNR/SSIM competitive. Cosmos remains strong in novel scenes given its broad pretraining, and EVAC is competitive under viewpoint shifts due to its image-space action encoding. IRASim degrades under distribution shift, likely because it trains solely on in-domain data with end-effector pose action representations.

BridgeV2W benefits jointly from pretrained vision and motion priors and pixel-space mask actions that accommodate diverse camera viewpoints. By preserving those priors during adaptation and aligning the action control signal with the pixel space in which pretrained video models operate, BridgeV2W achieves better generalization to novel scenes (RQ2) and camera viewpoints (RQ3) without trading off per-frame fidelity.

**Evaluation on AgiBot-G1.** Table 2 demonstrates that BridgeV2W achieves the best overall performance on the dual-arm AgiBot-G1 dataset: it attains lower FVD/LPIPS and higher Mask-IoU, while keeping PSNR / SSIM better. These results indicate that a unified world model design, conditioned by pixel-aligned embodiment masks, scales from single-arm to dual-arm embodiments without re-designing action encoders. It maintains temporal realism and action–video consistency despite increased kinematic complexity. These findings support RQ4 (a unified, cross-embodiment world model) while remaining compatible with the generalization observations made on DROID.

**Ablation Studies.** Table 3 shows that each component of BridgeV2W contributes to the video generation performance and robustness. Discarding pretraining (Row 1) weakens overall quality and stability under distribution shift, consistent with RQ2 on inheriting priors. Replacing pixel-aligned mask actions with end-effector pose coordinate-space actions (Row 2) yields notably worse temporal/perceptual quality and lower Mask-IoU, especially under unseen cameras and scenes, supporting RQ1 that mask actions better match the pixel-space distribution of pretrained video generators and are more view-aware. Removing the ControlNet-style conditioning (Row 3) further degrades results, indicating the value of a dedicated branch for injecting actions without eroding pretrained knowledge. Dropping the flow-based motion loss (Row 4) primarily reduces Mask-IoU and temporal coherence, aligning with our goal to emphasize dynamic, task-relevant regions rather than static background reconstruction. Taken together, the full design achieves the most reliable video generation quality and action-to-video consistency across all splits.

Table 3: Ablation study on different design choices of BridgeV2W.

| Model Variant | PSNR ↑ | SSIM ↑ | LPIPS ↓ | FVD ↓ | Mask-IoU ↑ |
|---|---|---|---|---|---|
| | In-Domain / Unseen Cam / Unseen Scene | | | | |
| w/o Pretrained Model | 21.24 / 19.61 / 17.72 | 0.842 / 0.770 / 0.568 | 0.120 / 0.145 / 0.209 | 211.3 / 245.3 / 513.6 | 59.7 / **56.1** / 34.2 |
| w/o Mask Action | 21.77 / 17.59 / 17.96 | 0.863 / 0.698 / 0.645 | 0.119 / 0.173 / 0.178 | 175.8 / 360.4 / 453.0 | 58.9 / 47.0 / 41.4 |
| w/o ControlNet | 21.38 / 19.57 / 17.17 | 0.819 / 0.716 / 0.687 | 0.122 / 0.165 / 0.192 | 194.9 / 255.2 / 446.2 | 58.9 / 53.4 / 36.8 |
| w/o Flow Loss | 20.72 / 20.02 / 18.98 | 0.859 / 0.826 / 0.705 | 0.132 / 0.146 / 0.153 | 201.4 / 235.7 / 420.4 | 58.3 / 52.1 / 39.7 |
| BridgeV2W (Ours) | **22.89 / 20.87 / 19.73** | **0.874 / 0.833 / 0.717** | **0.111 / 0.127 / 0.138** | **145.2 / 191.3 / 362.1** | **62.2** / 55.3 / **44.1** |

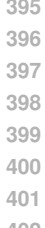
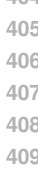
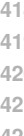

Figure 3: Qualitative video generation results produced by BridgeV2W.

**Qualitative Analysis.** Figure 3 demonstrates three key effects of BridgeV2W. (1) Pixel-space supervision on action masks transfers to unseen camera viewpoints, because the supervision is view-aligned rather than tied to a specific frame. (2) A pretrained video generation model enables synthesis in novel scenes, maintaining action-consistent motion despite background changes. (3) The unified mask representation serves as an embodiment-agnostic action representation, supporting a unified embodied world model across different robot platforms. More qualitative results of BridgeV2W performing in diverse scenarios could be found in Appendix D.

### 4.3 DOWNSTREAM APPLICATIONS

**Real-World Setup.** Real-world experiments use a Franka Research 3 robotic arm and a fixed third-person Zed2i RGB-D camera. For a given target gripper action, we employ frankapy (Zhang et al., 2020) for trajectory generation and feedback control to reach the target. We evaluate four tasks: put the bottle on the plate, put the toy in the shelf, close the drawer, and flip the cup, each with approximately 80 trajectories collected using a 3D mouse. For each task, initial object poses are randomized, and a trial is successful when the condition is met. We further implement three VLA baselines: OpenVLA-OFT (Kim et al., 2025), $\pi_0$ (Black et al., 2024), and SpatialVLA (Qu et al., 2025). More details could be found in Appendix C.1.

**Policy Evaluation.** We evaluate BridgeV2W as a proxy for policy evaluation. Given an initial image, we feed it to each VLA policy, take the resulting action chunk, and generate future images with BridgeV2W. We then use the last generated image to re-query the VLA models in an auto-regressive manner. The correlation between BridgeV2W-based evaluation and real-world success rates is shown in Figure 4. We observe a strong correlation (Pearson $r = 0.84$), indicating that BridgeV2W can serve as a proxy for estimating VLA policy performance without real-world execution. We also note that BridgeV2W sometimes predicts higher success rates than those observed in the real world, because BridgeV2W is trained primarily on successful expert demonstrations, it tends to generate successful rollouts when action errors are modest, whereas such errors can still cause failures in the real world.

Detailed results are provided in Table 4. We report two metrics: Mean Maximum Rank Violation (MMRV), adopted from SIMPLER (Li et al., 2025b), which measures the consistency between real-world and BridgeV2W policy rankings, and the Pearson correlation coefficient (Pearson $r$), which quantifies the linear relationship between the two evaluations. BridgeV2W performs well on both metrics, supporting its use as a proxy for downstream policy evaluation. Additional details are provided in Appendix C.2.

**Goal-Image Conditioned Manipulation.** We further apply BridgeV2W to goal-conditioned manipulation. For each task, we specify two goal images (a grasping subgoal and a final-state subgoal) and optimize action sequences with the Cross-Entropy Method (CEM). Concretely, at each iteration over the rollout horizon, we sample multiple action trajectories from a Gaussian distribution, use BridgeV2W to predict the resulting final states, score each trajectory by its similarity to the target goal image, select the top-$k$ trajectories as elites, and update the distribution's mean and variance accordingly. The executed action is taken as the mean of the refined distribution. Detailed results are reported in Table 5. We observe strong performance on tasks where the gripper remains down-facing with moderate rotation. For rotation-heavy tasks (e.g., closing a drawer, flipping a cup), the main difficulty stems not from mask limitations but from searching over high-dimensional rotational controls. Simple planning refinements, such as allocating more CEM samples to rotation and briefly decoupling rotation from translation, lead to clear success-rate gains. Additional analysis is provided in Appendix C.3, which also discusses how BridgeV2W can be integrated with VLA policies for closed-loop planning.

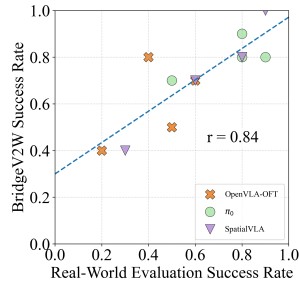

Figure 4: Correlation between BridgeV2W policy evaluation and real-world success rates.

Table 4: Policy evaluation across tasks and baselines using BridgeV2W.

| Tasks | MMRV ↓ | Pearson $r$ ↑ |
|---|---|---|
| Put on plate | 0.133 | 0.655 |
| Put in shelf | 0.033 | 0.944 |
| Close Drawer | 0.000 | 0.982 |
| Flip Cip | 0.033 | 0.945 |
| Avg. | 0.050 | 0.882 |

Table 5: Real-world planning performance with BridgeV2W.

| Tasks | OpenVLA-OFT | BridgeV2W |
|---|---|---|
| Put on plate | 4 / 10 | 5 / 10 |
| Put in shelf | 6 / 10 | 5 / 10 |
| Close Drawer | 5 / 10 | 5 / 10 |
| Flip Cip | 2 / 10 | 3 / 10 |
| All Tasks | 17 / 40 | 18 / 40 |

## 5 CONCLUSION

In this paper, we have proposed BridgeV2W, which turns actions into pixel-aligned embodiment masks (via URDF and camera parameters) and injects them into pretrained video generation models with a ControlNet-style pathway, supplemented by a motion-centric flow loss. This alignment preserves pretrained visual and motion priors, yields viewpoint robustness, and offers a unified world model architecture across diverse embodiments. We have further evaluated BridgeV2W's video generation capabilities on two robotic platforms and demonstrated its utility on downstream tasks, including policy evaluation and goal-conditioned manipulation.

**Limitations and Future Work.** BridgeV2W inherits typical video generation model failure modes such as hallucinations and long-horizon drift, and downstream applications may suffer from slow inference speed due to a heavy backbone. We plan to broaden pretraining with datasets lacking URDF or action labels (e.g., HOI) by deriving embodiment masks via segmentation, and to pursue a smaller, faster world model via distillation or lightweight adapters.

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

APPENDIX

## A   THE USE OF LARGE LANGUAGE MODELS (LLMS)

We use large language models only for language editing: rewriting sentences for clarity and concision, correcting grammar, harmonizing style, and suggesting alternative phrasings. The LLMs do not contribute to research ideation, problem formulation, dataset collection, experimental design, implementation, analysis, or the drafting of technical content.

## B   VIDEO GENERATION EXPERIMENTS

### B.1   DATASETS

We evaluate BridgeV2W on two robotic datasets: DROID (Khazatsky et al., 2024) (single-arm Franka) and AgiBot-G1 (Bu et al., 2025) (dual-arm). For training, we use 19k DROID trajectories, each recorded from two calibrated third-person cameras. We treat the two views as independent view–trajectory pairs sharing the same action stream, yielding $19\text{k} \times 2 = 38\text{k}$ view–trajectory pairs. For AgiBot-G1, we train on 15k trajectories and include all available calibrated external views per scene. For evaluation, we hold out 200 trajectories from each dataset as a standard test set with no sequence overlap with training. In addition, for DROID, we construct two generalization splits: an unseen-camera-viewpoint set of 100 trajectories (cameras not used in training) and an unseen-scene set of 100 trajectories (novel layouts/skills).

For the single-arm DROID setting, each action at time $t$ is a 7-D end-effector command $\mathbf{a}_t^{\text{(single)}} = [x_t, y_t, z_t, \phi_t, \theta_t, \psi_t, g_t] \in \mathbb{R}^7$, where $(x_t, y_t, z_t)$ is the Cartesian position, $(\phi_t, \theta_t, \psi_t)$ are roll, pitch, and yaw, and $g_t \in [0, 1]$ is the normalized gripper control. For the dual-arm AgiBot-G1 setting, we concatenate left- and right-arm commands, $\mathbf{a}_t^{\text{(dual)}} = [\mathbf{a}_t^{(L)}; \mathbf{a}_t^{(R)}] \in \mathbb{R}^{14}$, $\mathbf{a}_t^{(L)} = [x_t^{(L)}, y_t^{(L)}, z_t^{(L)}, \phi_t^{(L)}, \theta_t^{(L)}, \psi_t^{(L)}, g_t^{(L)}]$, $\mathbf{a}_t^{(R)} = [x_t^{(R)}, y_t^{(R)}, z_t^{(R)}, \phi_t^{(R)}, \theta_t^{(R)}, \psi_t^{(R)}, g_t^{(R)}]$, enabling synchronized bimanual control within the same pixel-aligned mask space.

Figure 5 visualizes two robot models from URDFs using Viser (Yi et al., 2025). For each action, we convert the Cartesian end-effector command (*xyz*+*rpy*+*gripper*) to joint space via inverse kinematics, apply forward kinematics on the URDF to obtain link poses, and project the meshes onto the image plane with calibrated intrinsics/extrinsics to render a pixel-aligned embodiment mask.

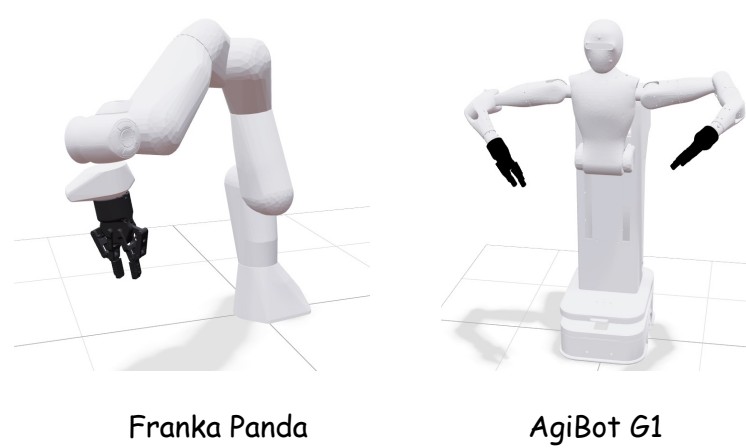

Franka Panda          AgiBot G1

Figure 5: The visualization of the URDF of Franka Panda and AgiBot G1.

## B.2 IMPLEMENTATION DETAILS

We adopt CogVideoX-5B-I2V (Yang et al., 2025c) as the backbone video generator (see Table 6 for model parameters). The text branch uses a T5 encoder (Raffel et al., 2020). Because our embodied world model conditions on action masks rather than language, we do not use the datasets' language instructions; instead, we supply a neutral placeholder prompt, *"Follow this action sequence represented in mask"*, to keep the text encoder active.

| Parameter | Value |
|---|---|
| number of layers | 42 |
| attention heads | 48 |
| hidden size | 3072 |
| positional encoding | RoPE |
| time embedding size | 256 |
| text length | 226 |
| max sequence length | 82k |
| training precision | BF16 |

Table 6: Hyper-Parameters of CogVideo-5b-I2V.

We inject embodiment masks as action conditions into the pretrained CogVideoX via a ControlNet-style conditioning branch. Fine-tuning hyperparameters are listed in Table 7. We fine-tune on $2 \times 8$ H20 GPUs (96 GB each) for approximately two weeks.

## B.3 BASELINES

**IRASim** (Zhu et al., 2024)     IRASim formulates robot world modeling as a trajectory to video prediction problem given $h$ historical observations $I_{t-h:t}$ and an action sequence $a_{t:t+n}$, the model predicts the next $n+1$ frames $I_{t+1:t+n+1} = f(I_{t-h:t}, a_{t:t+n})$. For efficiency, diffusion is performed in a VAE latent space: each frame is encoded/decoded by a frozen VAE, and a DiT backbone with memory-efficient spatio-temporal attention scales to long, high-resolution rollouts. To preserve visual context, historical frames are injected as *clean* (noise-free) tokens, while only future tokens are noised and supervised, allowing predictions to attend to clean history for temporal consistency. Action conditioning is implemented via adaptive layer normalization at two granularities: (i) *video-level* conditioning encodes the whole trajectory into a single embedding that modulates all blocks; and (ii) *frame-level* conditioning linearly encodes each action step, combines it with the diffusion-timestep embedding, and produces per-frame scale and shift parameters for spatial blocks (with a shared temporal-block embedding), explicitly enforcing action and enabling fine-grained robot-

| Parameter | Value |
|---|---|
| video size | (480, 720) |
| video length | 25 |
| $\lambda_{dyn}$ | 0.1 |
| dynamic window size $K$ | 4 |
| $\lambda_{flow}$ | 0.05 |
| $e_{switch}$ | 5 |
| learning rate | 2e-5 |
| batch size | 5 |
| optimizer | *adamw* |
| optimizer beta1 | 0.9 |
| optimizer beta2 | 0.95 |
| optimizer epsilon | 1e-8 |
| optimizer weight_decay | 1e-4 |
| max rad norm | 1.0 |
| lr scheduler | *constant with warmup* |
| lr warmup step | 100 |
| lora rank | 64 |
| lora alpha | 64 |

Table 7: Hyper-Parameters of BridgeV2W.

object interactions. A current limitation is that IRASim is trained only on in-domain data, which may hinder generalization to unseen cameras and scenes.

**Cosmos** (Agarwal et al., 2025)    Cosmos frames a world foundation model (WFM) as a learnable, at-scale digital twin for physical AI. The pipeline has two stages: (i) pre-train a generalist visual WFM on a large curated video corpus with scalable transformers and learned video tokenizers that compress videos into compact token sequences; (ii) post-train on target environments with prompt–video pairs where prompts encode action commands, trajectories, or instructions to obtain specialized, action-conditioned models. The model predicts future observations conditioned on executed actions, tightening the coupling between representation and controllable dynamics. While Cosmos benefits from pretraining to generalize to novel scenes, its end-effector pose action representation constrains transfer to unseen camera viewpoints and diverse embodiments.

**EVAC** (Jiang et al., 2025)    EVAC is an action-conditioned world model that predicts future visual observations from an agent's proposed actions, serving as a controllable, low-cost proxy for real-robot or 3D-simulator evaluation. It injects control signals via a multi-level conditioning scheme that combines end-effector projection action maps with delta-action encodings. To support multi-view robotics, EVAC encodes camera pose and motion using spatial cross-attention and ray-direction map embeddings, enabling view-consistent synthesis as viewpoints change. By projecting actions into the image plane, EVAC is robust to unseen camera viewpoints; however, because it is trained only on in-domain data and does not leverage pretrained visual or motion priors, its performance degrades in unseen scenes.

### B.4    ABLATION SETTINGS

In our ablation study, we probe four design choices of BridgeV2W while keeping all other factors fixed.

- *w/o Pretrained Model*: We keep the CogVideoX architecture but train the transformer backbone from scratch (no pretrained weights); the T5 text encoder and VAE remain pretrained. This isolates the contribution of pretrained visual and motion priors to scene generalization, texture fidelity, and rollout stability.

- *w/o Mask Action*: We replace mask-based actions with end-effector pose actions (e.g., $xyz + rpy + gripper$), encoded by an MLP. This ablation tests whether pixel-aligned supervision is the main driver of viewpoint robustness and cross-embodiment conditioning.

- *w/o ControlNet*: We remove the ControlNet-style pathway for mask conditioning and instead route conditions through a separate non-zero-initialized branch. This examines the role of a zero-initialized residual path in preserving pretrained priors during fine-tuning.

- *w/o Flow Loss*: We drop the motion-centric flow loss and train with the remaining objectives, to evaluate whether emphasizing task-relevant motion regions improves contact timing, motion sharpness, and background stability.

### B.5 TRAINING WITH SEGMENTATION-DERIVED MASKS AND REDUCED CALIBRATION REQUIREMENTS

A notable advantage of BridgeV2W is that it does not require camera intrinsics/extrinsics or URDF models during training. The framework only uses this geometric information at inference time to render pixel-aligned embodiment masks ("calc masks"). During training, the action-conditioned supervision can be obtained directly from segmentation-derived masks ("seg masks"), allowing the use of large uncalibrated datasets where no robot model or camera parameters are available.

To demonstrate this flexibility, we conducted additional experiments combining AgiBot-G1 robot data with the Ego4D FHO subset (Grauman et al., 2022), which contains egocentric human–hand interaction videos without any calibration or articulated models. We extracted hand masks using GroundedSAM and trained BridgeV2W under several data configurations. As shown in Table 8, training solely with segmentation-derived masks on robot data ("100% G1 seg") still provides meaningful supervisory signals despite the mismatch between segmentation masks and the calc masks used at inference. More importantly, incorporating large-scale Ego4D segmentation masks while mixing in only a small fraction of G1 calc-mask data nearly matches the performance of full calc-mask training. This indicates that: (i) segmentation-only human videos contribute rich motion priors, and (ii) limited calibrated robot data is sufficient for aligning the model with the target embodiment.

At inference time, BridgeV2W requires approximate camera parameters and a URDF model to render embodiment masks. This requirement is lightweight and practical: such metadata is routinely available in modern robotic systems for control, teleoperation, and simulation. Furthermore, the geometry used in our experiments is not derived from ideal checkerboard calibration. For example, many camera intrinsics/extrinsics in the DROID dataset come from CtRNet-X (Lu et al., 2025), a learned 2D–3D alignment pipeline that introduces estimation noise. Despite this approximation, BridgeV2W maintains strong performance across all benchmarks. Additionally, recent approaches such as URDFormer (Chen et al., 2024) can recover articulated robot models directly from RGB images when a URDF is not available.

Overall, these results show that BridgeV2W can scale to broad uncalibrated video corpora during training while requiring only lightweight geometric information during deployment. This combination enables both practical scalability and strong action-conditioned video generation performance.

Table 8: Training with segmentation-derived masks and mixed human–robot data.

| Data Source | PSNR ↑ | SSIM ↑ | LPIPS ↓ | FVD ↓ | Mask-IoU ↑ |
|---|---|---|---|---|---|
| 100% G1 **calc mask** (URDF-rendered) | 24.49 | **0.868** | **0.102** | 129.5 | **58.3** |
| 100% G1 **seg mask** (GroundedSAM) | 22.87 | 0.822 | 0.129 | 191.6 | 53.9 |
| 30% G1 **calc** + Ego4D **seg** | 24.28 | 0.850 | 0.118 | 133.9 | 57.2 |
| 70% G1 **seg** + 30% G1 **calc** + Ego4D **seg** | **24.58** | 0.863 | 0.108 | **118.5** | 58.1 |

### B.6 EVALUATING MASK-IOU AND THE ROLE OF GEOMETRIC SIGNALS

Mask-IoU is used to evaluate action–video consistency, which standard appearance metrics (PSNR/SSIM/LPIPS/FVD) cannot measure. Although BridgeV2W conditions on rendered masks, Mask-IoU compares predictions against *ground-truth* masks containing object interactions and occlusions, not against the conditioning inputs. Thus, simply receiving masks does not inflate the metric; indeed, ablations that keep the same masks but remove key modeling components yield lower Mask-IoU, showing that the metric reflects accurate motion prediction.

To test whether baselines could similarly benefit from geometry, we provide ground-truth camera intrinsics and extrinsics to baseline world models via MLP encoders. As shown in Table 9, calibration features do not improve performance and degrade unseen-camera results due to distribution shift.

Table 9: Effect of providing camera intrinsics/extrinsics to baseline world models.

| Methods | PSNR↑ | SSIM↑ | LPIPS↓ | FVD↓ | Mask-IoU↑ |
|---|---|---|---|---|---|
| IRASim | 22.11/18.02/16.23 | 0.846/0.763/0.672 | 0.119/0.162/0.166 | 175.7/415.8/583.8 | 58.0/45.9/32.7 |
| IRASim (with cam params) | 21.98/17.64/15.91 | 0.839/0.751/0.663 | 0.124/0.171/0.178 | 182.3/438.6/605.2 | 57.3/44.1/31.8 |
| Cosmos | 21.13/19.73/19.38 | 0.826/0.786/0.709 | 0.122/0.177/0.147 | 184.3/303.1/412.2 | 59.2/48.0/37.0 |
| Cosmos (with cam params) | 21.27/18.12/17.75 | 0.818/0.772/0.695 | 0.128/0.189/0.159 | 190.4/336.7/451.9 | 58.5/45.8/34.9 |
| BridgeV2W | 22.89/20.87/19.73 | 0.874/0.833/0.717 | 0.111/0.127/0.138 | 145.2/191.3/362.1 | 62.2/55.3/44.1 |

## C  REAL-WORLD EXPERIMENTS

### C.1  REAL-WORLD SETUP

**Robot Platform**    As shown in Fig. 6, our real-world platform consists of a statically mounted *Franka Research 3* robotic arm and a fixed third-person *Zed2i* RGB-D camera observing the workspace. The camera is rigidly mounted on a tripod facing the tabletop and provides synchronized RGB-D streams with timestamps. We calibrate the camera to robot extrinsics using a checkerboard, and use factory intrinsics for the *Zed2i*. For control, we employ `frankapy` (Zhang et al., 2020) to execute Cartesian pose trajectories with closed-loop feedback. To facilitate teleoperation during data collection, we use a 3D mouse to specify waypoints that `frankapy` interpolates into smooth, time-parameterized trajectories.

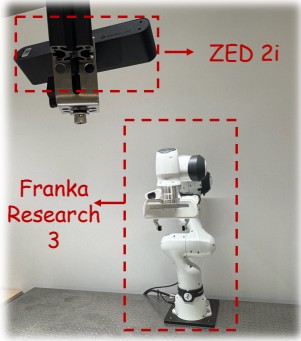

Figure 6: Illustrations of the real-world Franka robot platform setup.

**Task Definition**    As shown in Figure 7, We evaluate four manipulation tasks: (1) *Put the redbull on the plate (put on plate)*; (2) *Put the lion on the top shelf (put in shelf)*; (3) *Close the upper drawer (close drawer)*; (4) *Flip the cup upside down (flip cup)*. For each task, we collect approximately 80 teleoperated trajectories, yielding diverse object poses and initial scene layouts. Each episode starts from a standardized reset configuration with objects placed within the marked workspace. Success criteria are task-specific and binary (e.g., bottle resting on the plate without toppling; drawer fully closed; cup rotated to the target orientation).

**Vision-Language-Action (VLA) model baselines**    We consider three representative VLA baselines in our real-world evaluations:

- OpenVLA-OFT (Kim et al., 2025): An optimized fine-tuning recipe for VLA models that targets both control quality and inference speed. Using OpenVLA as the base policy, the method systematically rethinks three adaptation choices: parallel decoding action chunking, continuous vectorized control space, and a simple L1 regression over continuous actions.
- $\pi_0$ (Black et al., 2024): A generalist VLA policy that augments a pretrained vision–language model with a flow-matching action head to model continuous control. The architecture couples

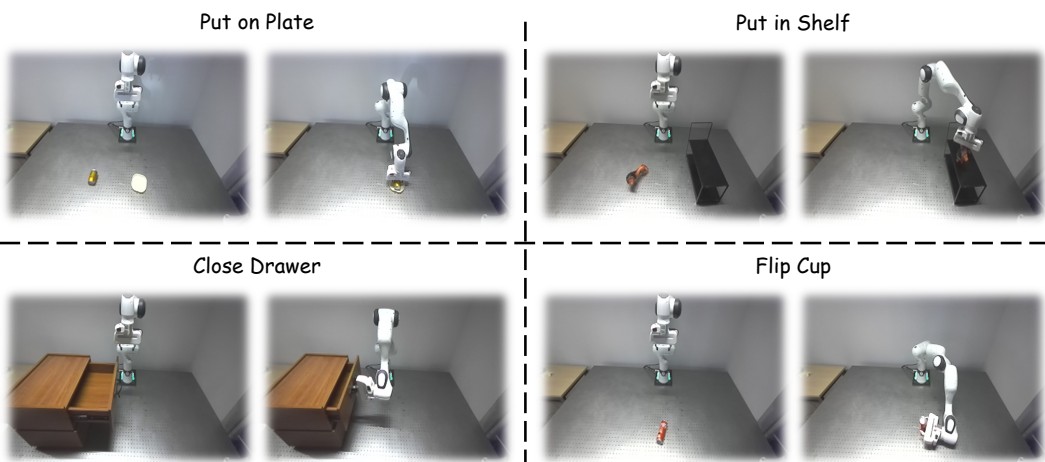

Figure 7: Illustrations of the real-world task setup.

Internet-scale semantic priors from the VLM with an action chunking decoder that outputs short horizons of continuous actions, enabling high-frequency control (up to ∼50 Hz).

- SpatialVLA (Qu et al., 2025): A VLA backbone with 3D spatial awareness via two representations: ego3D position encoding, which injects egocentric 3D cues into visual tokens; and adaptive action grids, which discretize continuous robot motions into data-driven spatial grids and learn action tokens aligned to the workspace geometry.

## C.2 POLICY EVALUATION

**Evaluation Procedure.** Starting from the initial observation $x_0$ (with optional instruction $c$), we evaluate each VLA policy $\pi$ by simulating its closed-loop behavior with BridgeV2W in an autoregressive manner. At step $t$, the policy consumes the current input image $x_t$ and outputs an action chunk $\mathbf{a}_{t:t+K-1} = \pi(x_t, c)$. Each action in the chunk is mapped to a pixel-aligned embodiment mask via $\phi$, yielding a mask sequence $m_{t:t+K-1} = \phi(\mathbf{a}_{t:t+K-1})$. The world model $\mathcal{W}$ then predicts a *video clip* conditioned on $x_t$ and the mask sequence,

$$\hat{\mathbf{x}}_{t+1:t+K} = \mathcal{W}(x_t, m_{t:t+K-1}),$$

where $\hat{\mathbf{x}}_{t+1:t+K}$ denotes $K$ consecutive future frames. We take the *last* frame of this clip, $\hat{x}_{t+K}$, as the next policy input by setting $x_{t+K} \leftarrow \hat{x}_{t+K}$, and continue the rollout by re-querying $\pi$ to obtain the next action chunk. This loop proceeds until a horizon $H$ is reached, producing a synthetic trajectory $\{\hat{\mathbf{x}}_{t+1:t+K}\}_{t=0}^{H-1}$ that approximates how the scene would evolve under each policy.

**Implementation Details.** BridgeV2W is trained with an action chunk length of 25. For VLA baselines whose chunk length is shorter than 25, we pad the sequence by repeating the final action until it reaches length 25. Task success is assessed by human raters who judge the BridgeV2W-generated videos. As future work, we plan to incorporate large multimodal models (e.g., ChatGPT (Achiam et al., 2023), Qwen (Yang et al., 2025a)) to automate the evaluation of generated rollouts.

**Metrics** Let $R = \{R_i\}_{i=1}^N$ denote real-world success rates and $R_S = \{R_{S,i}\}_{i=1}^N$ denote BridgeV2W success rates for the same $N$ policies. We adopt two metrics to evaluate the effectiveness of BridgeV2W as a proxy for policy evaluation:

(1) Mean Maximum Rank Violation (MMRV) (Li et al., 2025b) quantifies how severely the proxy misorders policies:

$$\text{RankViolation}(i, j) = |R_i - R_j| \cdot \mathbb{1}\left[(R_{S,i} < R_{S,j}) \neq (R_i < R_j)\right], \quad (8)$$

$$\text{MMRV}(R, R_S) = \frac{1}{N} \sum_{i=1}^{N} \max_{1 \leq j \leq N} \text{RankViolation}(i, j). \quad (9)$$

Here $\mathbb{1}[\cdot]$ is the indicator function; a larger MMRV means worse ranking consistency between BridgeV2W and real-world performance.

(2) Pearson correlation coefficient (Pearson $r$), which measures the linear association between $R$ and $R_S$:

$$r = \frac{\sum_{i=1}^{N}(R_i - \bar{R})(R_{S,i} - \bar{R}_S)}{\sqrt{\sum_{i=1}^{N}(R_i - \bar{R})^2}\sqrt{\sum_{i=1}^{N}(R_{S,i} - \bar{R}_S)^2}}, \quad \bar{R} = \frac{1}{N}\sum_{i=1}^{N}R_i, \ \bar{R}_S = \frac{1}{N}\sum_{i=1}^{N}R_{S,i}. \quad (10)$$

Values close to 1 indicate that higher real-world success corresponds to proportionally higher BridgeV2W success.

**Failure Cases** As summarized in Figure 4, BridgeV2W tends to overestimate success relative to real-world evaluations. To examine this bias, we conduct a case study illustrated in Figure 8: the left panel shows the correct action sequence from $\pi_0$; in the middle and right panels we inject $y$-axis offsets of $-0.02\,\mathrm{m}$ and $-0.10\,\mathrm{m}$, respectively. BridgeV2W still predicts a successful grasp under small perturbations, whereas such deviations cause failure on the real robot. We attribute this to training predominantly on expert demonstrations, which encourages the model to "recover" minor errors. Incorporating imperfect and failure trajectories during training (e.g., explicit negatives or perturbation-based augmentation) may improve alignment with real-world policy evaluation.

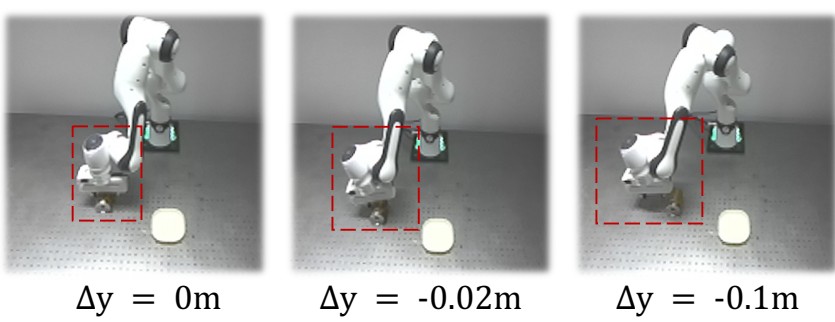

$$\Delta y = 0m \qquad \Delta y = -0.02m \qquad \Delta y = -0.1m$$

Figure 8: Case study of the failure cases in policy evaluation.

### C.3 GOAL-IMAGE CONDITIONED MANIPULATION.

**Planning Algorithm.** Following V-JEPA 2 (Assran et al., 2025) settings, we adopt the Cross-Entropy Method (CEM) within a Model Predictive Control (MPC) loop to plan. The detailed algorithm is illustrated in Algorithm 1. Given the current RGB frame $x_0$, the robot pose $p_0 = [x, y, z, r, p, y, g]$, a goal frame $x^{\mathrm{goal}}$, horizon $H$, world model $\mathcal{W}$, and CEM hyperparameters (iterations $T$, samples per iteration $S$, elites $k$), we maintain, for each step $h \in \{1, \ldots, H\}$, a factorized sampling distribution over 7-DoF actions: a Gaussian $\mathcal{N}(\mu_h, \Sigma_h)$ on pose *increments* $\Delta q_h \in \mathbb{R}^6$ (translation $xyz$, rotation $rpy$), and a Bernoulli parameter $\pi_h \in (0, 1)$ for the gripper $g_h \in \{0, 1\}$. At the start of each MPC cycle we initialize $\{(\mu_h, \Sigma_h, \pi_h)\}_{h=1}^{H}$ and run $T$ CEM refinements. In each refinement, we draw $S$ candidate sequences $A^{(s)} = \{a_h^{(s)}\}_{h=1}^{H}$ by sampling $\Delta q_h^{(s)} \sim \mathcal{N}(\mu_h, \Sigma_h)$ and $g_h^{(s)} \sim \mathrm{Bernoulli}(\pi_h)$, then forming $a_h^{(s)} = [\Delta q_{h,1:3}^{(s)}, \Delta q_{h,4:6}^{(s)}, g_h^{(s)}]$. Each candidate is rolled out with the world model to obtain the terminal prediction

$$x_{H+1}^{(s)} = \mathcal{W}(x_0, A^{(s)}),$$

and scored in a VAE latent space using mean $\ell_1$ distance. Let $\mathrm{Enc}(\cdot)$ be the encoder and $d$ the latent dimension; define

$$\mathcal{L}^{(s)} = \ell(z_{H+1}^{(s)}, z^{\mathrm{goal}}), \qquad z_{H+1}^{(s)} = \mathrm{Enc}(x_{H+1}^{(s)}), \ z^{\mathrm{goal}} = \mathrm{Enc}(x^{\mathrm{goal}}), \ \ell(u, v) = \frac{1}{d}\|u - v\|_1.$$

We select the $k$ elites with the lowest losses and refit each step's distributions: $(\mu_h, \Sigma_h)$ to the empirical mean/covariance of $\{\Delta q_h^{(s)}\}_{s \in I}$ and $\pi_h = \frac{1}{k}\sum_{s \in I}\mathbb{1}\{g_h^{(s)} = 1\}$. After $T$ refinements,

we extract the receding-horizon plan by taking per-step means and a $0.5$ threshold for the gripper,

$$A_h^\star = [\mu_h(1{:}3),\, \mu_h(4{:}6),\, \mathbb{1}\{\pi_h \geq 0.5\}], \quad h = 1, \ldots, H,$$

execute only the first control $a_1^\star = A_1^\star$, observe the new $(x_0, p_0)$, and re-plan with a shifted horizon until the goal is reached or a time limit is met.

**Implementation Details.** Each task is evaluated over 10 trials per method (VLA baselines and BridgeV2W). We discretize the translational action space with a step of $0.05\,\mathrm{m}$ along each axis, and the rotational space with a step of $20°$ (RPY). For simple pick-and-place tasks, we restrict the search to translation only and set rotational deltas to zero. Gripper commands are determined by a simple heuristic (open during approach, close at grasp, release at the target) without additional search.

**Planning Results Using BridgeV2W vs. VLA Baselines** Table 10 reports goal-image–based manipulation planning results. BridgeV2W attains strong performance on pick-and-place tasks but lags on tasks that require substantial rotations, likely because searching rotational subspaces is challenging for the world model during planning.

Table 10: BridgeV2W for goal-conditioned planning

| Tasks | OpenVLA-OFT | $\pi_0$ | SpatialVLA | BridgeV2W |
|---|---|---|---|---|
| Put on plate | 4 / 10 | 8 / 10 | 9 / 10 | 5 / 10 |
| Put in shelf | 6 / 10 | 9 / 10 | 8 / 10 | 5 / 10 |
| Close Drawer | 5 / 10 | 8 / 10 | 6 / 10 | 5 / 10 |
| Flip Cip | 2 / 10 | 7 / 10 | 7 / 10 | 3 / 10 |
| All Tasks | 17 / 40 | 32 / 40 | 30 / 40 | 18 / 40 |

**Qualitative Results and Failure Cases** Figure 9 presents goal-conditioned planning rollouts generated by BridgeV2W. In the two pick-and-place tasks (*put on plate* and *put in shelf*), the planner finds feasible action sequences without extensive exploration of rotational subspaces. In *close drawer*, the arm fails to execute the wrist rotation needed to approach and pull the handle, leading to a stall near the drawer front. In *flip cup*, the robot successfully grasps the cup but does not achieve the target orientation before placement, yielding an incorrect final pose.

**Analysis of Rotation-Heavy Manipulation Tasks** Rotation-intensive tasks introduce a significantly more complex action-search landscape than translation-dominated pick-and-place settings. In these tasks, the gripper must rotate around multiple axes while maintaining contact consistency, leading to a higher-dimensional and highly non-convex optimization space for Cross-Entropy Method (CEM) planning.

Despite the inherent difficulty of these tasks, BridgeV2W itself remains stable under large wrist rotations. As shown in Table 11, metrics such as LPIPS, FVD, and Mask-IoU do not degrade for rotation-heavy tasks, and the predicted trajectories remain visually coherent. This indicates that the world model captures the necessary rotational dynamics and that mask conditioning does not collapse under self-occlusion or viewpoint changes. Our experiments reveal that the dominant source of error arises from the action-search process rather than the world model. Naïve CEM search distributes samples uniformly across translation and rotation, causing the planner to under-explore the rotational subspace. This leads to failure cases where the robot reaches the correct position but does not complete the required rotation.

We evaluate simple refinements to the search procedure within the same BridgeV2W model:

- **Rotation-first search**: temporarily freeze translation while optimizing rotation.
- **Sample reallocation**: increase the number of CEM samples dedicated to rotational dimensions.
- **Axis-wise decomposition**: search yaw, pitch, and roll components sequentially.

These strategies yield clear improvements on rotation-heavy tasks: success rates increase from 3/10 to 5/10 for *close drawer* and from 0/10 to 3/10 for *flip cup*. Importantly, these gains are achieved without modifying BridgeV2W, indicating that the model already encodes sufficient geometry and dynamics for rotation reasoning.

Table 11: Model quality metrics and real-world success rates on manipulation tasks.

| Task | PSNR↑ | SSIM↑ | LPIPS↓ | FVD↓ | Mask-IoU↑ |
|------|-------|-------|--------|------|-----------|
| Put on plate | 26.25 | 0.906 | 0.097 | 179.0 | 63.8 |
| Put in shelf | 26.17 | 0.914 | 0.098 | 190.1 | 61.4 |
| Close drawer | 26.42 | 0.925 | 0.091 | 175.6 | 64.9 |
| Flip cup | 26.02 | 0.901 | 0.103 | 180.9 | 62.1 |

**Integration with VLA Frameworks for Closed-Loop Planning**    BridgeV2W can be used as a predictive critic within modern vision–language–action (VLA) frameworks, enabling efficient closed-loop planning. To examine this capability, we conducted preliminary real-world experiments in which OpenVLA-OFT served as the action proposer. At each control step, OpenVLA-OFT generated ten action rollouts by varying its decoding temperature, and BridgeV2W predicted the visual outcomes of each rollout. The rollout whose predicted outcome best matched the task objective was selected and executed.

This two-stage OpenVLA-OFT + BridgeV2W framework yields consistent improvements over OpenVLA alone across multiple manipulation tasks (Table 12). Because VLA models generate structured and semantically coherent action hypotheses, this closed-loop paradigm is substantially more search-efficient than offline CEM, which must sample blindly in the continuous action space. The primary limitation is increased inference latency due to evaluating multiple rollouts per step. Potential remedies include distilling BridgeV2W into a lighter predictive model, adding a small learned value head for faster scoring, or caching intermediate video features for incremental rollout evaluation.

Table 12: Real-world success rates when integrating BridgeV2W into an OpenVLA-OFT closed-loop planning pipeline.

| Task | OpenVLA-OFT | OpenVLA + BridgeV2W |
|------|-------------|---------------------|
| Put on plate | 4/10 | 6/10 |
| Put in shelf | 6/10 | 7/10 |
| Close drawer | 5/10 | 5/10 |
| Flip cup | 2/10 | 4/10 |
| All Tasks | 17/40 | **22/40** |

# D   ADDITIONAL VISUALIZATION RESULTS

This section presents additional video generation examples produced by **BridgeV2W**. Figure 10 shows in-domain DROID rollouts (top row) alongside the same action sequences rendered from unseen camera viewpoints (bottom row), highlighting viewpoint robustness. Figure 11 illustrates DROID scenes under unseen backgrounds to assess generalization beyond the training distribution. Figure 12 reports examples on the AgiBot G1 dataset to demonstrate cross-embodiment applicability.

---

**Algorithm 1:** CEM–MPC planning with BridgeV2W

---

**Input:** Current frame $x_0$, current pose $p_0 = [x, y, z, r, p, y, g]$ (Euler xyz + gripper);
Goal frame $x^{\text{goal}}$; world model $\mathcal{W}$; horizon $H$;
CEM iterations $T$, VAE Encoder Enc, samples $S$, elites $k$;
Init per-step Gaussian $(\mu_h, \Sigma_h)$ over $\Delta q_h \in \mathbb{R}^6$ $(xyz, rpy)$, and Bernoulli parameter
$\pi_h \in (0, 1)$ for $g_h \in \{0, 1\}$;
Objective $\ell(\cdot, \cdot)$ (default mean $L_1$ on latent space).
**Output:** First control $a_1^\star$ and (optionally) plan $A_{1:H}^\star$.

1 **repeat**
  // 1) CEM initialization
2  initialize $\{\mu_h, \Sigma_h, \pi_h\}_{h=1}^{H}$
3  **for** $t \leftarrow 1$ **to** $T$ **do**
4   **for** $s \leftarrow 1$ **to** $S$ **do**
5    $x_1^{(s)} \leftarrow x_0,\ p_1^{(s)} \leftarrow p_0,\ A^{(s)} \leftarrow \emptyset$
6    **for** $h \leftarrow 1$ **to** $H$ **do**
     // Sample 7-DoF action: $xyz + rpy + gripper$
7     $\Delta q_h \sim \mathcal{N}(\mu_h, \Sigma_h),\quad g_h \sim \text{Bernoulli}(\pi_h)$
8     $a_h \leftarrow [\Delta q_{h,1:3},\ \Delta q_{h,4:6},\ g_h]$
9     Append $a_h$ to $A^{(s)}$
10    **end**
11    $x_{H+1}^{(s)} \leftarrow \mathcal{W}(x_0, A^{(s)})$
12    $\mathcal{L}^{(s)} \leftarrow \ell(\text{Enc}(x_{H+1}^{(s)}), \text{Enc}(x^{\text{goal}}))$
13   **end**
   // 2) Select elites and refit distributions
14  $I \leftarrow \text{argsort}(\{\mathcal{L}^{(s)}\})[: k]$
15  **for** $h \leftarrow 1$ **to** $H$ **do**
16   Fit $\mu_h, \Sigma_h$ to $\{\Delta q_h^{(s)}\}_{s \in I}$
17   $\pi_h \leftarrow \frac{1}{k} \sum_{s \in I} \mathbb{1}\{g_h^{(s)} = 1\}$
18  **end**
19  **end**
  // 3) Extract plan & execute
20  $A_h^\star \leftarrow [\mu_h(1{:}3),\ \mu_h(4{:}6),\ \mathbb{1}\{\pi_h \geq 0.5\}]\quad$ for $h = 1..H$
21  Apply $a_1^\star \leftarrow A_1^\star$, observe new $(x_0, p_0)$, recede horizon
22 **until** *goal reached* **or** *time limit*

---

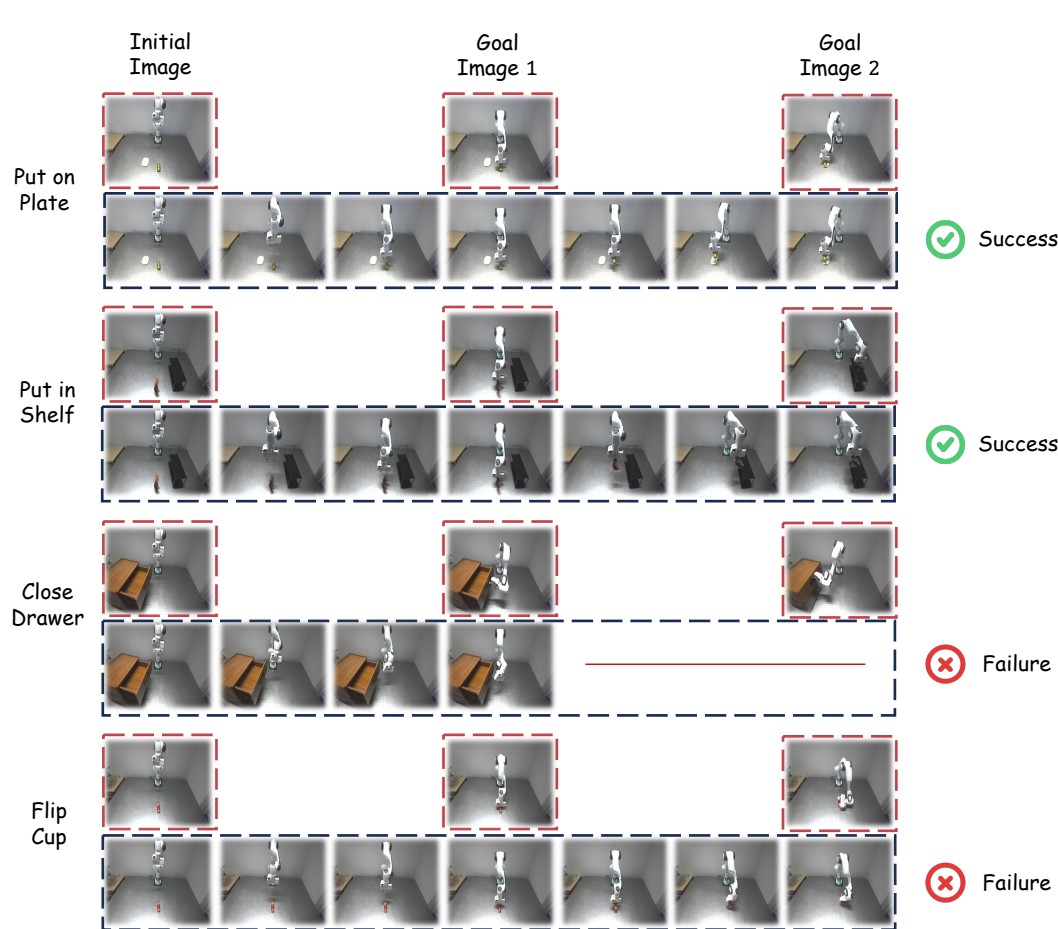

Figure 9: Qualitative rollouts for goal-conditioned planning with BridgeV2W. Top row: initial observation and two goal images. Bottom row: model-predicted execution frames.

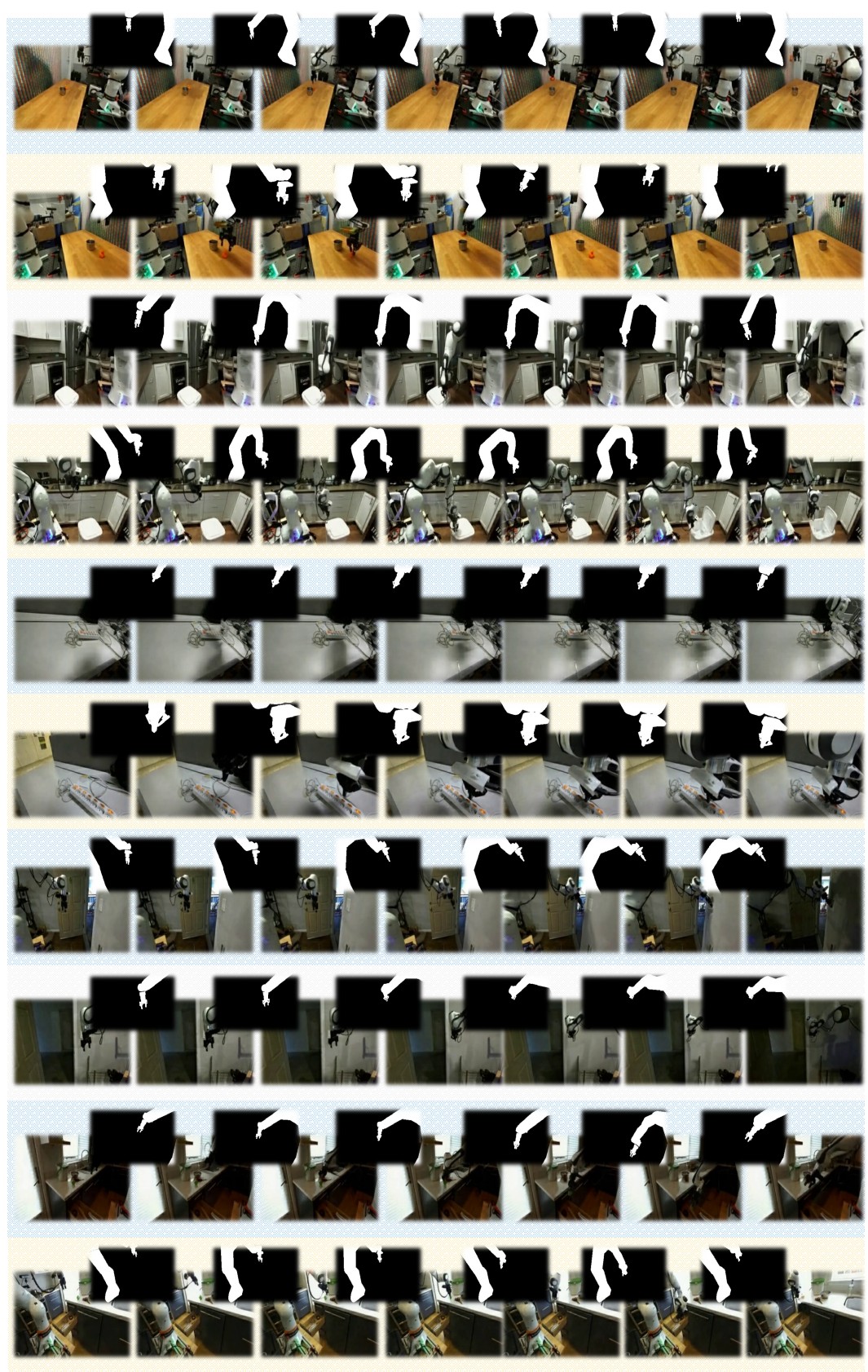

Figure 10: Visualization results in DROID of in-domain camera and unseen camera.

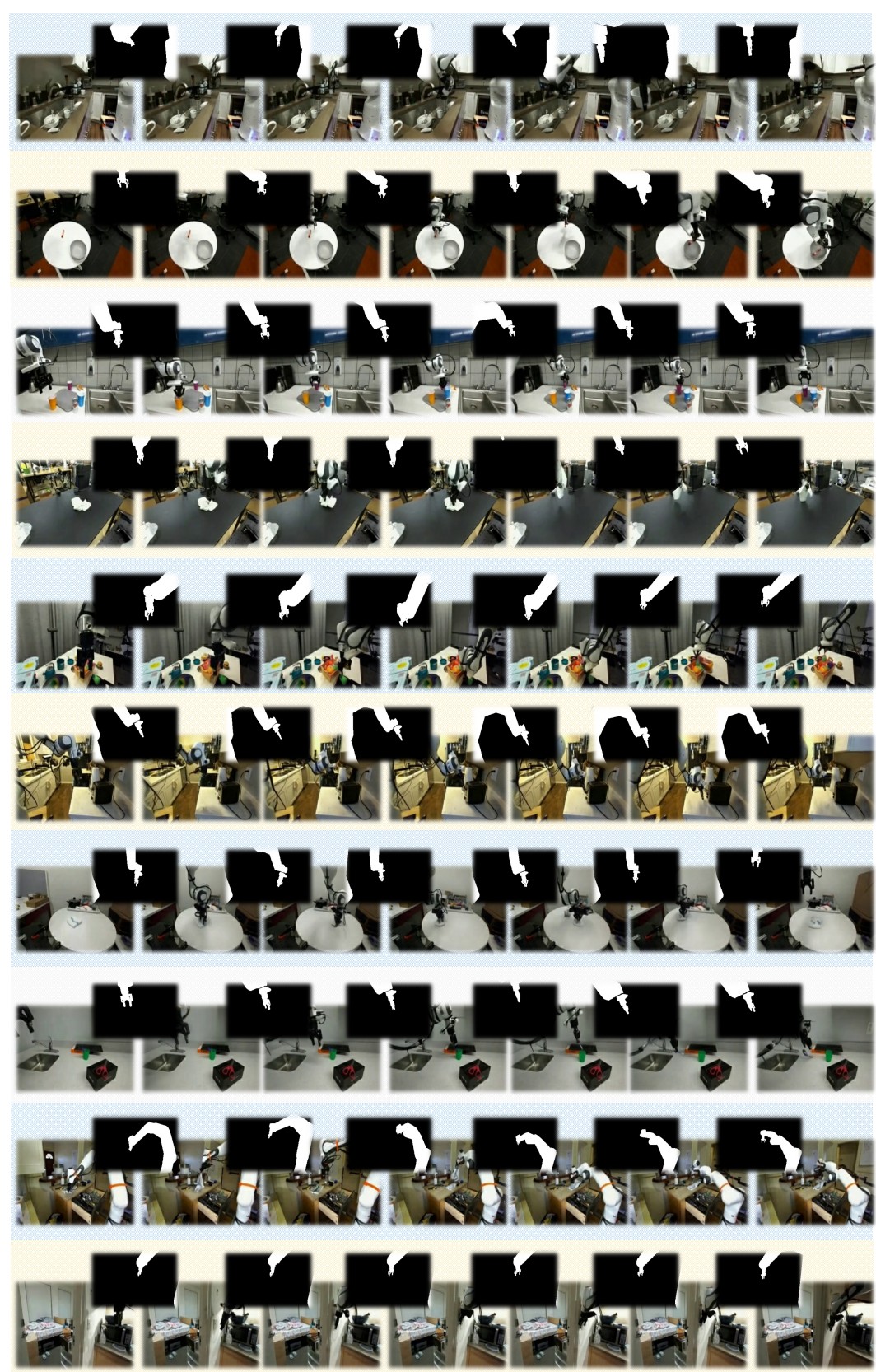

Figure 11: Visualization results in DROID of unseen scenes.

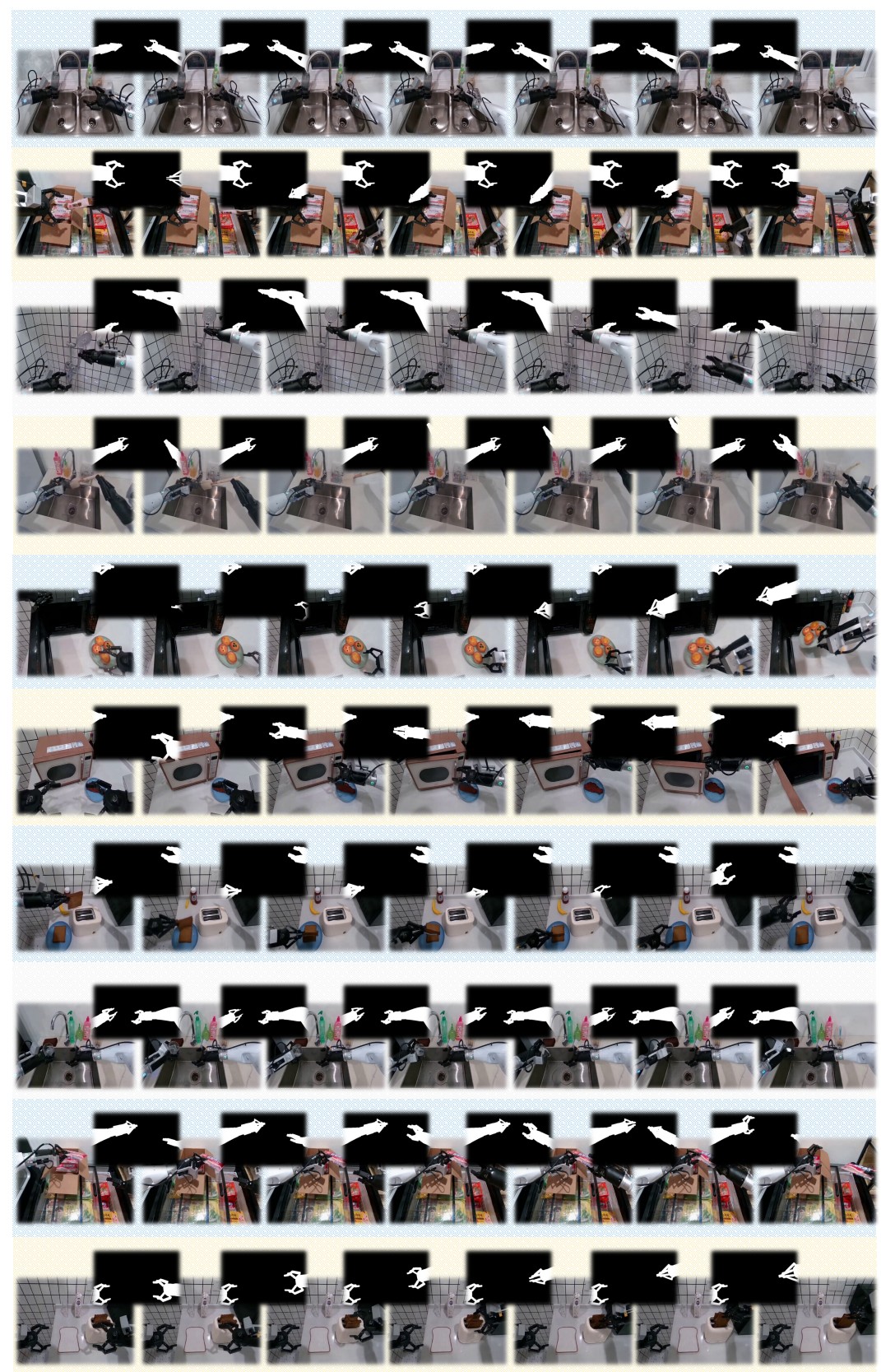

Figure 12: Visualization results in AgiBot G1.

