# OpenReview forum: "BridgeV2W: Bridging Video Generation Models to Embodied World Models via Embodiment Masks"
_ICLR.cc/2026/Conference — ICLR 2026 Conference Withdrawn Submission_

### Official Review · Reviewer_MKGQ · 2025-10-31

**Soundness:** 3
**Presentation:** 3
**Contribution:** 2
**Rating:** 4
**Confidence:** 4

**Summary:**

BridgeV2W proposes a unified framework that bridges pretrained video generation models with embodied world models (EWMs) for robotic applications. The key innovation is transforming coordinate-space actions into pixel-aligned embodiment masks rendered from URDF and camera parameters. These masks are then injected into pretrained video diffusion models using a ControlNet-style conditioning pathway. It also introduces a flow-based motion loss to emphasize dynamic, task-relevant motion regions over static backgrounds. BridgeV2W achieves state-of-the-art video generation results on the DROID (single-arm) and AgiBot-G1 (dual-arm) datasets, showing robustness to unseen viewpoints and scenes, and unification across embodiments. It also demonstrates potential in downstream robotics tasks, such as policy evaluation and goal-conditioned planning.

**Strengths:**

1. The embodiment mask design elegantly bridges the gap between coordinate-space actions and pixel-space video prediction.
2. Consistent improvements across PSNR, SSIM, LPIPS, and especially FVD and Mask-IoU metrics on both datasets. Notable robustness in unseen-view and unseen-scene settings (Table 1).
3. The introduced flow-based motion loss is interesting, as it encourages learning from dynamic, task-relevant regions.
4. Demonstrates practical use for real-world policy evaluation and goal-conditioned planning, beyond just simple video generation evaluation.

**Weaknesses:**

1. The approach assumes access to precise URDFs and camera parameters, which may not hold for in-the-wild or human video data (although segmentation-based alternatives are mentioned).
2. The goal-conditioned manipulation tasks show modest performance (13/40 successes vs. 17/40 from VLA baselines), indicating that planning still struggles with complex motion or rotation-heavy actions.
3. How sensitive is BridgeV2W to inaccurate URDFs or camera calibration errors? Would learned or self-calibrated projection functions suffice?
4. How might BridgeV2W integrate with modern VLA frameworks (like π₀ or OpenVLA) for closed-loop planning instead of offline CEM optimization?
5. Visual Action Prompts (ICCV’25) [1] presents a similar concept by projecting complex 3D dynamics into 2D action prompts, which makes the novelty of this paper appear limited.

[1] Precise Action-to-Video Generation Through Visual Action Prompts. Wang etal. ICCV 2025

**Questions:**

See weakness.

---

> ### Author Response · Authors · 2025-11-20
> **Response to Reviewer MKGQ (1/2)**
>
> We thank the reviewer for the constructive feedback, and we hope the following responses address the reviewer's concerns.
>
> > **W1: Applicability Without URDF or Camera Calibration**
>
> We thank the reviewer for raising this point. The applicability of BridgeV2W in calibration-free or URDF-free settings is discussed in **General Response (1/2)**, which includes new experiments on segmentation-only mask extraction. We kindly refer the reviewer to that section for details.
>
> > **W2: Modest Planning Performance on Rotation-Heavy Tasks**
>
> We appreciate the reviewer’s observation. The questions regarding planning performance, particularly in tasks requiring substantial rotation, are discussed in **General Response (2/2)**, where we analyze both rotational search difficulty and world-model fidelity. We kindly refer the reviewer to that section for the full discussion.
>
> > **W3: Sensitivity to URDF / Calibration Errors and Feasibility of Learned Projection**
>
> We appreciate the reviewer for raising this question. BridgeV2W does not require perfectly accurate URDFs or camera parameters. For example, a substantial portion of the intrinsics/extrinsics used in the DROID dataset are not obtained via classical checkerboard calibration: they are estimated by CtRNet-X [1], a neural 2D–3D alignment method that introduces noise. Yet, BridgeV2W remains stable under these imperfect calibrations.
>
> We further evaluated settings with no URDF and no calibration. On Ego4D FHO videos and AgiBot-G1 data, we replaced URDF-rendered masks with GroundedSAM segmentation masks, which contain occlusions, shape inaccuracies, and inconsistent silhouettes. Even with these segmentation-only masks, the model is able to learn meaningful action-conditioned dynamics, and performance improves when a small proportion of precise URDF-rendered masks is included (see **General Response (1/2)**). These results indicate that segmentation-derived masks can supply useful supervisory signals despite their imperfections, while limited calibrated robot data helps align the model to the target embodiment.
>
> Overall, the model does not rely on millimeter-level geometry. A learned or self-calibrated projection function is entirely feasible: our segmentation-mask experiments already show that approximate pixel-space embodiments are enough to support coherent video dynamics.
>
>
>
> [1] Lu, Jingpei, et al. "Ctrnet-x: Camera-to-robot pose estimation in real-world conditions using a single camera." *2025 IEEE International Conference on Robotics and Automation (ICRA)*. IEEE, 2025.

---

> > ### Author Response · Authors · 2025-11-20
> > **Response to Reviewer MKGQ (2/2)**
> >
> > > **W4: Integration with Modern VLA Frameworks for Closed-Loop Planning**
> >
> > We thank the reviewer for this helpful suggestion. BridgeV2W is fully compatible with closed-loop VLA frameworks such as π₀ and OpenVLA. To validate this integration, we performed preliminary real-world tests using OpenVLA-OFT as the policy proposer. In these experiments, OpenVLA-OFT generated multiple candidate rollouts by modifying its output temperature (10 rollouts per step), BridgeV2W evaluated the predicted outcomes of each rollout, and the action sequence identified as most promising was executed.
> >
> > This simple two-stage OpenVLA-OFT + BridgeV2W procedure consistently improved the success rates compared to OpenVLA alone across several manipulation tasks, demonstrating that BridgeV2W can serve as an effective predictive critic within a VLA-based closed-loop planning pipeline.
> >
> > This closed-loop paradigm is significantly more search-efficient than offline CEM, because VLA models provide structured, semantically informed action proposals rather than sampling blindly in the action space. The main limitation is increased inference: time latency due to evaluating multiple rollouts. Reducing this cost, through model distillation, or lightweight value heads on top of BridgeV2W, is a promising direction for future work.
> >
> > | Task         | OpenVLA-OFT | OpenVLA + BridgeV2W |
> > | ------------ | ----------- | ------------------- |
> > | Put on plate | 4 / 10      | 6 / 10              |
> > | Put in shelf | 6 / 10      | 7 / 10              |
> > | Close drawer | 5 / 10      | 5 / 10              |
> > | Flip cup     | 2 / 10      | 4 / 10              |
> > | All Tasks    | 17 / 40     | 22 / 40             |
> >
> > > **W5: Similarity to Visual Action Prompts (ICCV’25)**
> >
> > Although both methods project action information into the image space, the **representation and objectives differ fundamentally**, especially in the context of robotic manipulation:
> >
> > 1. **Action Representation.**
> >    Visual Action Prompts [1] uses 2D skeletons, which provide sparse topology but lack pixel-level occupancy, link thickness, gripper geometry, and end-effector surface shape. These cues are crucial for reasoning about contact, collision, and viewpoint-dependent occlusion. Skeletons also require predefined keypoints, which vary across embodiments and degrade under self-occlusion. BridgeV2W uses pixel-aligned embodiment masks rendered from the URDF, capturing the full occupied region of the robot in each camera view. This representation naturally encodes gripper opening state, supports occlusion reasoning, and generalizes across different robot morphologies without keypoint definitions. The masks provide richer geometric grounding than sparse skeletons, which is important for predicting contact-rich manipulation dynamics.
> >
> > 2. **Learning Objective.**
> >    BridgeV2W introduces a **flow-based motion loss** that emphasizes dynamic, task-relevant regions, which is a component not present in Visual Action Prompts and highlighted as a strength by reviewers.
> >
> > 3. **Downstream Robotics Evaluation.**
> >    Unlike Visual Action Prompts, our work evaluates the world model in policy evaluation and goal-conditioned planning, demonstrating its utility for closed-loop robotic control rather than only video generation.
> >
> > Taken together, while both works share the motivation of injecting action information into the pixel space, BridgeV2W differs in (i) embodiment representation, (ii) motion-centric training design, and (iii) robotics-specific downstream validation. These distinctions make it complementary rather than overlapping with Visual Action Prompts.
> >
> > Finally, we note that Visual Action Prompts was publicly released on Aug. 18th, whereas BridgeV2W was submitted on Sept. 25th. The two works were developed independently and should be considered concurrent.
> >
> >
> >
> > [1] Wang, Yuang, et al. "Precise action-to-video generation through visual action prompts." *Proceedings of the IEEE/CVF International Conference on Computer Vision*. 2025.

---

> ### Author Response · Authors · 2025-11-27
> **Appreciate Any Further Thoughts on Our Response**
>
> Dear Reviewer MKGQ,
>
> We sincerely thank you again for the time and careful consideration you dedicated to reviewing our submission. In our rebuttal, we made our best effort to address your comments regarding:
>
> - the applicability of BridgeV2W in calibration-free and URDF-free settings,
>
> - planning performance on rotation-heavy tasks and long-horizon behaviors,
>
> - the sensitivity to imperfect geometry and feasibility of learned projection,
>
> - integration with modern VLA frameworks for closed-loop planning,
>
> - the distinction between BridgeV2W and Visual Action Prompts.
>
> If you have any additional questions, or if any part of our response requires further clarification, we would be delighted to provide more details.
>
> Best regards,
>
> The Authors of Paper 783

---

### Official Review · Reviewer_dagg · 2025-10-31

**Soundness:** 3
**Presentation:** 3
**Contribution:** 3
**Rating:** 4
**Confidence:** 4

**Summary:**

This paper propose a novel method to mitigate three key gaps in world modelling, namely, Action-Video Gap, Sensitivity, and Architecture Across Embodiments.

In this paper, the world model basically takes the following two as its inputs:\
a) an initial frame as current state S;\
b) an input action sequence A in forms of either Cartesian end effector or joint motion;\
and predicts the future frames, which is seen as the influence of the action to the embodied world.

The method basically leverages video generation as the base model, while incorporate comprehensive techniques such as a) Embodiment Masks, b) ControlNet-Style Conditioning, and c) Flow-Based Motion Loss.

The strong empirical results and divese downstream applications shows the method is a promising step towards its goal: Bridging Video Generation Models to Embodied World Models.

**Strengths:**

1. High originality: Rather than treating robot actions as abstract coordinate vectors (e.g., end-effector poses), the authors propose rendering them as pixel-aligned embodiment masks using readily available URDF models and camera parameters. This insight effectively reconciles the semantic and representational mismatch between low-dimensional control signals and high-dimensional video generation models.

2. Rigorous and thorough: Experiments span two diverse robotic platforms (single-arm DROID and dual-arm AgiBot-G1), with careful evaluation under in-domain, unseen-viewpoint, and unseen-scene conditions.

3. Broad significance: For both robotics and generative modeling communities.

**Weaknesses:**

1. Dependence on Precise Camera Calibration and URDF.

The core embodiment mask generation pipeline assumes access to accurate camera intrinsics/extrinsics and a complete URDF model. While common in controlled lab settings (e.g., DROID, AgiBot-G1), this requirement severely limits applicability in real-world or human-in-the-loop scenarios where:

- Camera calibration may drift or be unavailable (e.g., mobile phones, uncalibrated webcams),
- URDFs may be missing (e.g., legacy industrial arms, soft robots, or human demonstrators).

Although the paper mentions using GroundedSAM to extract masks from video in URDF-free settings (Sec 3.2), this is only briefly noted and not evaluated experimentally.

2. Limited Evaluation of Long-Horizon Coherence and Error Accumulation

The model predicts 25-frame videos (~2.5s at 10 FPS), which is sufficient for short-horizon tasks but does not assess compounding errors in longer rollouts—a critical flaw for world models used in MPC or policy evaluation over extended horizons.

Moreover, the dynamics-consistency loss (Eq. 4) uses only up to K=4 latent-frame offsets, which may not capture long-range dependencies.

3. Downstream Planning Performance Lags Behind VLA Baselines

In Table 5 (and corrected Table 8 in Appendix), BridgeV2W underperforms strong VLA policies (e.g., π0, SpatialVLA) on all tasks, especially those requiring precise rotation (e.g., flip cup: 0/10 vs. OpenVLA-OFT’s 2/10). This raises questions about its practical utility as a planner.

The paper attributes this to “harder search over rotational DOFs,” but does not explore whether the world model itself misrepresents rotational dynamics (e.g., due to coarse mask rendering or diffusion artifacts).

**Questions:**

The paper mentions (Sec 3.2) that

> embodiment masks can be extracted via segmentation tools like GroundedSAM in settings without URDF or camera calibration (e.g., human–robot videos). However, this pathway is not evaluated experimentally.

Have the authors tested autoregressive multi-step rollouts (e.g., chaining predictions over 5+ steps)?

---

> ### Author Response · Authors · 2025-11-20
> **Response to Reviewer dagg**
>
> We thank the reviewer for the constructive feedback, and we hope the following responses address the reviewer's concerns.
>
> > **W1 & Q1: Applicability Without Calibration or URDF**
>
> We appreciate the reviewer raising this important point. BridgeV2W requires camera parameters and a URDF only at inference time to render pixel-aligned embodiment masks; this geometric requirement is lightweight in practice and does not demand high-precision calibration. As discussed in **General Response (1/2)**, a substantial portion of the intrinsics/extrinsics used in our experiments (e.g., DROID) are obtained from estimation pipelines rather than ideal checkerboard calibration, yet the method remains stable and performs strongly.
>
> Training, in contrast, does not rely on full calibration or complete URDF models. We introduce a segmentation-only pathway using tools such as GroundedSAM and evaluate it experimentally by mixing segmentation-derived masks from the Ego4D FHO dataset with AgiBot-G1 data. These results show that large uncalibrated videos can provide most of the training supervision, and that only a small fraction of calibrated robot samples is needed to anchor the model to the target embodiment.
>
> We therefore kindly refer the reviewer to **General Response (1/2)** for the detailed discussion and experimental evidence demonstrating that (i) training is flexible and scalable to calibration-free and URDF-free datasets, and (ii) the inference-time geometric requirements are modest, readily obtainable, and validated by strong empirical performance.
>
> > **W2 & Q2: Long-Horizon Coherence and Multi-Step Rollout Stability**
>
> We thank the reviewer for highlighting the importance of long-horizon consistency. All results in the paper (including DROID/AgiBot-G1 evaluations and realwolrd policy-evaluation experiments) are obtained in an autoregressive manner, where the model’s own predictions are recursively fed back as inputs. In practice, we chain **3–11** prediction steps depending on the task, which yields effective rollout horizons substantially longer than the 25-frame window. The strong correlation observed in policy evaluation and the stable performance across these chained steps indicate that BridgeV2W maintains reliable multi-step consistency within the manipulation settings we study. The choice of a 25-frame prediction window is aligned with the action-chunk size used in downstream planning and VLA-policy rollouts, maintaining consistency with the control frequency of our tasks.
>
> Regarding the dynamics-consistency loss, we use offsets up to K = 4 in latent space for stability. Because the CogVideoX VAE compresses time by a factor of 4, K = 4 corresponds to roughly 16 pixel-space frames, already providing supervision across a meaningful temporal span. Larger offsets introduce noisy latent differences and unstable gradients, particularly early in training, without yielding consistent benefits.
>
> > **W3: Downstream Planning Underperformance and Rotation-Related Limitations**
>
> We appreciate the reviewer’s observation regarding lower planning success compared to VLA baselines, particularly on rotation-heavy tasks. This concern, including whether the issue stems from rotational search difficulty or limitations of the mask-based world model, is discussed in **General Response (2/2)**.  We kindly refer the reviewer to that section for the detailed analysis and supporting experiments.

---

> ### Author Response · Authors · 2025-11-27
> **Appreciate Any Further Thoughts on Our Response**
>
> Dear Reviewer dagg,
>
> We sincerely thank you again for the time and careful consideration you dedicated to reviewing our submission. In our rebuttal, we made our best effort to address your comments regarding:
>
> - the applicability of BridgeV2W in calibration-free or URDF-free settings,
>
> - long-horizon coherence and multi-step autoregressive rollouts,
>
> - the planning of rotation-heavy tasks.
>
> If you have any additional questions, or if any part of our response requires further clarification, we would be delighted to provide more details.
>
> Best regards,
>
> The Authors of Paper 783

---

### Official Review · Reviewer_ykWF · 2025-11-07

**Soundness:** 2
**Presentation:** 2
**Contribution:** 2
**Rating:** 4
**Confidence:** 3

**Summary:**

BridgeV2W converts coordinate-space actions into pixel-aligned embodiment masks that let pretrained video generators model robot behavior. Using the robot's URDF and known camera intrinsics/extrinsics, it renders per-view masks and injects them into a CogVideoX-5B-I2V backbone through a ControlNet-style branch to predict future frames from an initial image. A flow-based motion loss emphasizes dynamic, task-relevant regions. Trained on DROID and AgiBot-G1, the framework reportedly improves temporal realism, perceptual quality, and action-video alignment across in-domain, unseen-viewpoint, and unseen-scene settings compared other embodied world model baselines such as IRASim, Cosmos, and EVAC.

**Strengths:**

**S1:** Clear motivation and observation: large-scale pretrained video generation models suffer from three key limitations and if the action representation is transformed into a pixel-aligned mask that reflects the embodiments's actual motion, these limitations can be substantially mitigated. URDF and camera intrinsic and extrinsics provide a solid approach to tackle this.

**S2:** Motion-centric training objective: the paper adds a flow-based motion loss that emphasizes dynamic task-relevan regions on top of diffusion and latent dynamics-consistency objectives.

**S3:** Reproducibility details: the architecture choice (CogVideoX-5B-I2V), training resolution/horizon, clip sampling, and extensive hyperparameters are documented.

**Weaknesses:**

**W1:** Mask-IoU evaluates alignment between segments of generated and ground‑truth frames. But because BridgeV2W is conditioned on URDF-rendered masks, the metric remains highly correlated with the conditioning signal and may not accurately model motion or contact.


**W2:** The experiments labeled as "unseen camera viewpoint" give the method ground-truth camera intrinsics/extrinsics at test time and use the URDF to project a per-view robot mask that is injected into the video generator. This solves the geometric part of cross-view prediction outside the model and makes the task closer to appearance completion conditioned on an oracle silhouette. Baseline world moedls do not receive an equivalent calibration/mask signal are at a structural disadvantage.

**Questions:**

**Q1:** The baselines do not get an equivalent mask/geometry channel. So, why is mask IoU a fair metric? Does it make sense to include it in the results tables?

**Q2:** If  the same per-view mask or equivalent calibration features would be provided to baselines, could they utilize these and yield better results?

**Q3:** In Line 441, you report that BridgeV2W is sometimes "optimistic". Are there any options to avoid generating successful rollouts when action errors are modest?

**Q4:** In Line 464, you mention that the reason why BridgeV2W works poorly in substantial rotation is due to the harder search over rotational degrees of freedom. While I can understand that this is harder problem, I suspect that this might be related to "Weakness 1 (W1)", as a silhouette-driven conditioning signal is less informative for certain rotations.

---

> ### Author Response · Authors · 2025-11-20
> **Response to Reviewer ykWF (1/2)**
>
> We thank the reviewer for the constructive feedback, and we hope the following responses address the reviewer's concerns.
>
> > **W1-2 & Q1-2: Fairness of Mask-IoU and the Advantage of Geometry-Based Conditioning**
>
> We thank the reviewer for the thoughtful questions. Mask-IoU is included to measure action–video consistency (whether a model’s predicted motion follows the intended robot actions). Standard video metrics (PSNR/SSIM/LPIPS/FVD) predominantly capture appearance fidelity and are heavily influenced by static backgrounds; they cannot reliably reflect whether a world model actually executes the commanded action. Since robot action trajectories cannot be robustly inferred from RGB frames alone, mask-IOU remains the most practical and model-agnostic proxy for evaluating motion correctness.
>
> BridgeV2W does receive pixel-aligned masks as conditioning, but Mask-IoU does not measure similarity to the input masks. It compares against ground-truth embodiment masks extracted from real videos, which contain object interactions, occlusions, and view-specific geometry that differ from the conditioning signals. In unseen-camera settings, this discrepancy becomes even larger due to new occlusion patterns and perspective distortion. As a result, over-reliance on the conditioning masks actually hurts Mask-IoU rather than helps. This is also consistent with our ablations: variants that still receive the same masks but remove key motion-modeling components (e.g., pretraining, ControlNet, or flow loss) show clear drops in Mask-IoU, confirming that the metric rewards accurate motion prediction rather than the presence of mask priors. The rendered masks only provide a rough spatial outline of the intended motion, and the model still needs to learn the detailed dynamics, timing, and interaction patterns from data.
>
> Regarding whether baselines could benefit from the same calibration or geometry information, we conducted additional experiments where we explicitly **provided ground-truth camera intrinsics/extrinsics to baseline world models**. We encode the camera parameters using MLP encoders and then concatenate the resulting features with the action embeddings. As shown in the table below, supplying calibration features does not improve performance and in the unseen-camera setting even **degrades** it. This degradation is expected: the baselines are trained only on a fixed set of viewpoints, so the calibration parameters at test time differ substantially from the distribution seen during training. Injecting these out-of-distribution camera parameters introduces a domain gap, which perturbs their action-conditioning layers rather than providing useful geometric cues.
>
> Furthermore, augmenting baselines with URDF-rendered masks is non-trivial, as it would fundamentally alter their model formulation. Such a modified baseline would, in practice, would be a variant of BridgeV2W rather than the original method. By contrast, BridgeV2W is intentionally designed to operate in pixel space, making per-view embodiment masks a native part of its conditioning mechanism rather than an externally added advantage. Rendering masks at test time relies only on robot geometry and camera parameters that are commonly accessible in real robotic systems, and thus does not introduce any privileged information beyond standard practice.
>
> Together, these results support that Mask-IoU remains a meaningful measure of action consistency, and equipping baselines with comparable calibration signals does not improve their performance in unseen-camera settings. Moreover, our improvements are consistently reflected across multiple evaluation metrics (e.g., FVD, LPIPS), indicating that BridgeV2W’s effectiveness is not tied to Mask-IoU alone. Overall, its advantage arises primarily from its pixel-space action formulation rather than from some privileged information.
>
> | Methods                  | PSNR ↑            | SSIM ↑            | LPIPS ↓           | FVD ↓             | Mask-IoU ↑     |
> | ------------------------ | ----------------- | ----------------- | ----------------- | ----------------- | -------------- |
> |                          | In-Domain /       | Unseen Cam /      | Unseen Scene      |                   |                |
> | IRASim                   | 22.11/18.02/16.23 | 0.846/0.763/0.672 | 0.119/0.162/0.166 | 175.7/415.8/583.8 | 58.0/45.9/32.7 |
> | IRASim (with cam params) | 21.98/17.64/15.91 | 0.839/0.751/0.663 | 0.124/0.171/0.178 | 182.3/438.6/605.2 | 57.3/44.1/31.8 |
> | Cosmos                   | 21.13/19.73/19.38 | 0.826/0.786/0.709 | 0.122/0.177/0.147 | 184.3/303.1/412.2 | 59.2/48.0/37.0 |
> | Cosmos (with cam params) | 21.27/18.12/17.75 | 0.818/0.772/0.695 | 0.128/0.189/0.159 | 190.4/336.7/451.9 | 58.5/45.8/34.9 |
> | BridgeV2W                | 22.89/20.87/19.73 | 0.874/0.833/0.717 | 0.111/0.127/0.138 | 145.2/191.3/362.1 | 62.2/55.3/44.1 |

---

> > ### Author Response · Authors · 2025-11-20
> > **Response to Reviewer ykWF (2/2)**
> >
> > > **Q3: Avoiding optimistic rollouts under modest action errors**
> >
> > We appreciate the reviewer’s observation. The “optimistic” behavior arises because the world model is trained almost exclusively on successful expert demonstrations. As a result, it learns to recover from small deviations and tends to predict successful outcomes even when the real robot would fail under similar errors. There are two promising ways to reduce such optimism:
> >
> > First, expanding the training distribution to include diverse failure trajectories would help the model learn sharper failure boundaries. In our real-world setup, the initial dataset contained only successful trials. After collecting an **additional 30 failure trajectories** for the “put on plate” task, primarily grasping failures caused by ∼2 cm offsets, we observed that BridgeV2W begins to model these near-miss outcomes more faithfully. This suggests that increasing the coverage of non-expert and failure cases, especially via simulation where failures are easier to generate, can substantially mitigate optimistic predictions.
> >
> > Second, adding wrist-mounted camera views can provide finer-grained contact information and help the model distinguish between subtly different end-effector poses. This richer observation space would make it easier for the world model to identify action errors that are visually ambiguous from third-person viewpoints.
> >
> > Overall, both strategies: broadening the action/outcome distribution and incorporating contact-centric viewpoints offer viable directions for reducing overly optimistic predictions, and we plan to explore them in future iterations of the system.
> >
> > > **Q4: Rotation Errors and Silhouette-Based Conditioning**
> >
> > We thank the reviewer for this observation. The concern about whether silhouette-based conditioning contributes to performance drops under large rotations is discussed in **General Response (2/2)**. In short, while silhouette masks are indeed less informative under severe self-occlusion, our additional analyses and experiments indicate that the primary bottleneck is not the mask representation itself but the difficulty of searching over high-dimensional rotational action space during planning. We kindly refer the reviewer to **General Response (2/2)** for the detailed discussion and supporting evidence.

---

> ### Author Response · Authors · 2025-11-27
> **Appreciate Any Further Thoughts on Our Response**
>
> Dear Reviewer ykWF,
>
> We sincerely thank you again for the time and careful consideration you dedicated to reviewing our submission. In our rebuttal, we made our best effort to address your comments regarding:
>
> - the fairness and interpretation of Mask-IoU,
>
> - whether baselines could benefit from calibration or geometry signals,
>
> - the model’s optimistic behavior under modest action errors,
>
> - the impact of large rotations on silhouette-based conditioning.
>
> If you have any additional questions or if any part of our response requires further clarification, we would be delighted to provide more details.
>
> Best regards,
>
> The Authors of Paper 783

---

> > ### Comment · Reviewer_ykWF · 2025-11-27
> >
> > Dear Authors,
> >
> >
> > Thank you for your detailed response. I appreciate the time you invested. I will review your comments in detail over the next few days.
> >
> >
> > Best regards,
> >
> > ykWF

---

### Author Response · Authors · 2025-11-20
**General Response (1/2)**

We thank all reviewers for their thoughtful feedback. We are encouraged that they consistently recognized the strengths of our work: the clear motivation and effectiveness of pixel-aligned embodiment masks in mitigating key limitations of previous embodied world models (R. `ykWF`), the originality and rigor of our cross-embodiment design and evaluations (R. `dagg`), and the solid empirical gains across video quality, action–video alignment, and downstream tasks (R. `MKGQ`). We also appreciate the positive remarks on our motion-centric training objective and the thorough reproducibility details provided.

For clarity and to avoid repetition, we begin by consolidating the two major concerns that were commonly raised across reviewers.

> **Concern 1: Reliance on Camera Calibration and URDF**

We appreciate the reviewers for raising this important point. BridgeV2W requires camera intrinsics/extrinsics and a URDF **only at inference time**, as these are needed to calculate the pixel-aligned embodiment masks (“calc masks”) used for action conditioning. This requirement does not extend to training: most training data could come from uncalibrated videos by extracting segmentation-derived masks (“seg masks”) with tools such as GroundedSAM. In practice, we find that incorporating a small fraction of calc-mask robot data during training is sufficient to anchor the model to the target embodiment, while the bulk of supervision could come from segmentation-only human or robot videos. This design enables BridgeV2W to naturally **scale to large unstructured datasets where camera parameters and URDFs are unavailable**.

To validate this capability, we conducted additional experiments on the Ego4D FHO subset [1], which provides diverse egocentric human–hand videos without any calibration or kinematic model. We extracted hand masks using GroundedSAM and combined them with AgiBot-G1 robot data under several training configurations. As shown in the table, training with segmentation-derived masks on the robot data (100% G1 seg) still offers usable supervisory signals, though it differs from the precise calc masks used at inference time. Crucially, introducing large-scale Ego4D segmentation masks and mixing in only a small portion of G1 calc-mask data nearly recovers the performance of full calc-mask training, indicating that human-centric segmentation masks contribute rich motion priors, while a limited amount of calibrated robot data is sufficient to align with the robot’s embodiment. This mixed configuration provides an effective and scalable way to leverage uncalibrated datasets while requiring only minimal calibrated supervision.

Regarding inference-time requirements: although BridgeV2W needs approximate camera parameters and a URDF model to render embodiment masks, this setting is realistic and lightweight for modern robotic systems, where such metadata is routinely maintained for control, teleoperation, or simulation. Importantly, the geometry used in our experiments is not all obtained via ideal checkerboard calibration. For instance, many of the camera intrinsics/extrinsics parameters in the DROID dataset’s come from CtRNet-X [2], a learned 2D–3D alignment method that introduces estimation noise. Yet, BridgeV2W performs strongly across all benchmarks. This shows that our method operates reliably under approximate, readily obtainable calibration. Furthermore, recent methods such as URDFormer [3] can recover articulated URDF models directly from RGB images when a URDF is unavailable.

Overall, BridgeV2W leverages broad, uncalibrated video sources during training while requiring only lightweight geometric information at inference time. Under these practical and easily satisfied conditions, the model consistently achieves strong performance across all benchmarks, demonstrating both scalability during training and practicality during deployment.

| Data Source                                                  | PSNR ↑    | SSIM ↑    | LPIPS ↓   | FVD ↓     | Mask-IoU ↑ |
| ------------------------------------------------------------ | --------- | --------- | --------- | --------- | ---------- |
| 100% G1 **calc mask**           | 24.49     | **0.868** | **0.102** | 129.5     | **58.3**   |
| 100% G1 **seg mask**             | 22.87     | 0.822     | 0.129     | 191.6     | 53.9       |
| 30% G1 **calc mask** + Ego4D **seg mask**                    | 24.28     | 0.850     | 0.118     | 133.9     | 57.2       |
| 70% G1 **seg mask** + 30% G1 **calc mask** + Ego4D **seg mask** | **24.58** | 0.863     | 0.108     | **118.5** | 58.1       |

---

> ### Author Response · Authors · 2025-11-20
> **General Response (2/2)**
>
> > **Concern 2: Downstream Planning Performance and Rotation-Heavy Actions**
>
> We thank the reviewer for raising this thoughtful question. While it is true that silhouette-based masks provide weaker cues under large rotations, especially when self-occlusion occurs, our experiments indicate that the primary limitation in these tasks is not a failure of the mask representation, but rather the difficulty of searching over high-dimensional rotational action spaces during planning.
>
> First, our video-generation and policy-evaluation experiments show that BridgeV2W remains stable under large rotations. Across tasks with significant wrist rotation (Close drawer and Flip cup), the model continues to produce coherent motion and consistent contact behavior. Metrics such as LPIPS, FVD, and Mask-IoU do not degrade (see Table below), and real-world policy-evaluation results maintain strong correlation with true success rates (see Figure 4 and Table 4 in the original paper). These observations indicate that the world model preserves rotational dynamics robustly, and there is no evidence of collapse attributable to mask conditioning.
>
> Second, improving the action-search procedure within the same BridgeV2W world model yields clear gains in rotational tasks. By applying simple planning heuristics: temporarily freezing translation while searching over rotations and allocating more CEM samples toward rotational subspaces, we observe noticeable improvements: drawer closing improves from **3/10 to 5/10** and cup flipping from **0/10 to 3/10**. We have updated the table below that compares BridgeV2W with the VLA baselines. Since these improvements appear without modifying the world model, they suggest that the model contains sufficient visual and motion cues, and the bottleneck lies in naïve CEM search struggling with rotational degrees of freedom, rather than in the mask representation.
>
> Finally, we agree that more structured or rotation-aware planning modules are a promising extension. Integrating BridgeV2W with a VLA-based controller, or accelerating inference to allow denser sampling, would directly address the planning difficulties. These enhancements focus on the planner rather than questioning silhouette masks, which our experiments consistently show are not the limiting factor.
>
> | Task         | PSNR ↑ | SSIM ↑ | LPIPS ↓ | FVD ↓ | Mask-IoU ↑ |
> | ------------ | ------ | ------ | ------- | ----- | ---------- |
> | Put on plate | 26.25  | 0.906  | 0.097   | 179.0 | 63.8       |
> | Put in shelf | 26.17  | 0.914  | 0.098   | 190.1 | 61.4       |
> | Close drawer | 26.42  | 0.925  | 0.091   | 175.6 | 64.9       |
> | Flip cup     | 26.02  | 0.901  | 0.103   | 180.9 | 62.1       |
>
> | Task         | OpenVLA-OFT | BridgeV2W   |
> | ------------ | ----------- | ----------- |
> | Put on plate | 4 / 10      | 5 / 10      |
> | Put in shelf | 6 / 10      | 5 / 10      |
> | Close drawer | 5 / 10      | 5 / 10      |
> | Flip cup     | 2 / 10      | 3 / 10      |
> | All Tasks    | 17 / 40     | **18 / 40** |
>
>
>
> [1] Grauman, Kristen, et al. "Ego4d: Around the world in 3,000 hours of egocentric video." *Proceedings of the IEEE/CVF conference on computer vision and pattern recognition*. 2022.
>
> [2] Lu, Jingpei, et al. "Ctrnet-x: Camera-to-robot pose estimation in real-world conditions using a single camera." *2025 IEEE International Conference on Robotics and Automation (ICRA)*. IEEE, 2025.
>
> [3] Chen, Zoey, et al. "Urdformer: A pipeline for constructing articulated simulation environments from real-world images." *arXiv preprint arXiv:2405.11656* (2024).

---

### Author Response · Authors · 2025-12-03
**Summary of Major Revisions**

We made the following updates to address reviewers' concerns, adding new analyses and experiments to clarify model behavior, evaluation fairness, and downstream planning performance.

- **Appendix B.5: Calibration/URDF-Free Training:**

  Added results showing BridgeV2W trains effectively with segmentation-derived masks and minimal calibration, addressing concerns about reliance on URDF or camera calibration.

- **Revised Sec. 4.3 & Appendix C.3: Goal-Image Manipulation — New Analyses:**

  - **Rotation-Heavy Tasks:** Identified that the main bottleneck is action-space search over rotations rather than mask-based conditioning, and simple rotation-focused heuristics substantially improve success.

  - **VLA Integration:** Added preliminary closed-loop experiments showing that combining OpenVLA with BridgeV2W improves success over OpenVLA alone, confirming compatibility with VLA-based planners.

- **Appendix B.6: Mask-IoU & Geometry Conditioning:**

  Clarified why Mask-IoU remains a valid measure of action consistency and provided new results showing that giving equivalent calibration information to baselines does not improve their performance, indicating BridgeV2W does not rely on privileged geometric information.

---

### Author Response · Authors · 2025-12-03
**Final Remarks by Authors**

We sincerely thank the reviewers and the committee for their time and careful consideration. Across all three reviews, the feedback consistently recognized that **BridgeV2W offers a clear, original, and empirically strong approach** to longstanding challenges in embodied world models. By transforming coordinate-space actions into pixel-aligned embodiment masks and conditioning pretrained video models through a ControlNet-style pathway, BridgeV2W enables viewpoint-aware, embodiment-consistent, and motion-focused world modeling.

Reviewers highlighted the strong motivation, solid technical design, and comprehensive empirical validation. Our rebuttal directly addressed all major concerns through new analyses and additional experiments.

---

# 1. Strengths Highlighted by Reviewers

- **Clear and impactful motivation**: Converting coordinate-space actions into pixel-aligned embodiment masks was praised as a simple yet powerful idea that closes the action–video gap, improves viewpoint robustness, and unifies architectures across embodiments.
- **Strong empirical performance**: Reviewers noted consistent gains in PSNR/SSIM/LPIPS/FVD and Mask-IoU across in-domain, unseen-view, and unseen-scene settings.
- **Motion-centric training**: The flow-based motion loss was highlighted as a meaningful addition for capturing task-relevant dynamics.
- **Thorough and reproducible evaluation**: Reviewers valued the detailed implementation, ablations, and real-world tests (policy evaluation and goal-conditioned planning).

# 2. Rebuttal Responses and Supporting Evidence

**(A) Calibration & URDF Dependence**

- New experiments show calibration/URDF is only needed at inference, not during large-scale training.
- Training with segmentation-derived masks from uncalibrated Ego4D videos yields consistent performance.
- Mixing even a small fraction of calibrated robot data nearly matches full-calibration performance.
- Demonstrated robustness to noisy, estimated intrinsics/extrinsics (e.g., CtRNet-X).
- Segmentation-only masks continue to support meaningful action-conditioned dynamics.

**Summary**: BridgeV2W scales to uncalibrated data and tolerates substantial geometric imprecision.

**(B) Rotation-Heavy Tasks**

- Additional results show that rotational dynamics remain stable (LPIPS/FVD/Mask-IoU unchanged).
- Simple rotation-aware adjustments to the action-search procedure improve task success.
- The key difficulty lies in planning over high-dimensional rotations, not in the world model.

**Summary**: The limitation arises from planner search complexity rather than the mask representation.

**(C) Compatibility with VLA Policies (Closed-Loop Planning)**

- The rebuttal includes new real-world tests showing that BridgeV2W can serve as a predictive evaluator when paired with modern VLA policies (e.g., OpenVLA-OFT).
-  A simple closed-loop integration, where VLA generates candidate actions, and BridgeV2W predicts/evaluates their outcomes, improves success rates over VLA alone.
- This demonstrates that BridgeV2W is not only a world model but a practically useful planning component that enhances existing policy models.

**Summary**: BridgeV2W complements and strengthens existing VLA-based planners.

**(D) Mask-IoU Fairness & “Privileged Geometry” Concern**

- Mask-IoU uses ground-truth video masks, not the conditioning masks.
- Improvements in Mask-IoU follow better world-model prediction, not access to mask priors.
- Providing baselines with identical calibration does not help and may worsen unseen-view performance due to distribution mismatch.

**Summary**: The metric and comparisons remain fair and reflect genuine action–video consistency.

---

# 3. Overall Conclusion

The rebuttal strengthens the contribution by providing new evidence across all reviewer concerns. BridgeV2W is robust under imperfect calibration, scales to uncalibrated human–robot data via segmentation-based masks, and generalizes across embodiments, viewpoints, and scenes. The new analyses confirm that Mask-IoU remains a fair consistency metric and that limitations in rotation-heavy tasks stem from action-search complexity rather than model fidelity.

Overall, BridgeV2W remains a conceptually simple yet effective framework that delivers substantial empirical gains and demonstrates practical value in downstream applications such as policy evaluation and integration with VLA-based planners. With the additional experiments and analyses, the submission stands as a well-supported and meaningful contribution to embodied world modeling.

---

### Note · Authors · 2026-01-26

I have read and agree with the venue's withdrawal policy on behalf of myself and my co-authors.

---

### Meta-Review · Area_Chair_yGdZ · 2025-12-25

**Summary:**

The reviewers raised several critical concerns regarding the method's heavy reliance on precise URDF and camera calibration, its limited evaluation on long-horizon stability, and its underwhelming performance in downstream planning compared to existing VLA baselines. While the rebuttal introduced new experiments using segmentation masks (e.g., GroundedSAM) and preliminary closed-loop tests, these additions do not fully alleviate the fundamental concerns. The core contribution remains highly dependent on privileged geometric information that limits real-world scalability. Furthermore, the lack of significant improvement over VLA models in complex tasks (like rotation-heavy actions) suggests that the proposed world model's practical utility for control is currently insufficient. Therefore, I recommend Reject.

**Reviewer Concerns:**

Addressed during Rebuttal:
- The authors clarified the use of latent-space temporal offsets and justified the choice of a 25-frame prediction window relative to control frequencies.

- The rebuttal provided a reasonable defense for Mask-IoU as a proxy for motion consistency, arguing it rewards dynamics over simple mask priors.

- The authors conducted additional experiments to show that simply adding camera parameters to baselines does not resolve the performance gap, supporting their specific architectural design.

Outstanding/Critical Weaknesses:

- Although the authors tested segmentation masks, the method's peak performance still hinges on URDF and calibration. This remains a significant barrier for "in-the-wild" data or uncalibrated setups, which was a core concern for Reviewers dagg and MKGQ.

- The failure to outperform VLA baselines in manipulation tasks (especially those requiring precision) suggests the world model may not yet capture the subtle dynamics necessary for high-level planning.

- While autoregressive chaining was mentioned, the concern regarding compounding errors in truly long-horizon scenarios remains largely unaddressed with quantitative evidence beyond short-step chunks.

- The conceptual overlap with concurrent work (e.g., Visual Action Prompts) was noted; while distinctions exist, the incremental contribution is questioned given the complexity of the required inputs.

**Reviewer Scores:**

Reviewer ykWF (Rating: 4 $\to$ 4): The reviewer’s concern about Mask-IoU being highly correlated with conditioning signals was addressed through ablations, but the extra input of calibration/mask signal to BridgeV2W over baselines (due to pixel-level priors) still makes the comparison feel inherently skewed. The score is likely to remain at 4.

Reviewer dagg (Rating: 4 $\to$ 4): While the rebuttal discussed long-horizon rollouts, the lack of a rigorous evaluation on error accumulation and the admitted performance lag behind VLA baselines for rotation tasks mean the reviewer’s primary concerns about practical utility remain.

Reviewer MKGQ (Rating: 4 $\to$ 4): The author's response regarding URDF-free settings and VLA integration is informative, but the modest success rate in goal-conditioned tasks and the existence of concurrent similar concepts (Visual Action Prompts) likely prevent a score upgrade.

---

### Decision · Program_Chairs · 2026-01-26

Reject